

# Have you ever seen the rain? Observing a record convective rainfall with national and local monitoring networks and opportunistic sensors

Louise Petersson Wårdh[1,2*], Hasan Hosseini [1,2], Remco van de Beek[2], Jafet C.M. Andersson[2], Hossein Hashemi[1], Jonas Olsson[1,2]

[1] Division of Water Resources Engineering, Faculty of Engineering, Lund University, P.O. Box 118, 22100 Lund, Sweden

[2] Swedish Meteorological and Hydrological Institute (SMHI), Folkborgsvägen 17, Norrköping SE-601 76, Sweden

*Corresponding author. E-mail: louise.petersson_wardh@tvrl.lth.se.

## Abstract

Short-duration extreme rainfall can cause severe impacts in built environments and flood mitigation measures require high-resolution rainfall data to be effective. It is a particular challenge to observe convective storms which are expected to intensify with climate change. However, rainfall monitoring networks operated by national meteorological and hydrological services generally have limited ability to observe rainfall at sub-hourly and sub-kilometre scale. This paper investigates the capability of second- and third-party rainfall sensors to observe a highly localized convective storm that hit southwestern Sweden in August 2022. Specifically, we compared the observations from professional weather stations, C-band radar, X-band radar, Commercial Microwave Links and Personal Weather Stations to get a full impression of the sensors' strengths and weaknesses in the context of convective storms. The results suggest that second- and third-party networks can contribute with important information on short-duration extreme rainfall to national weather services. The second-party network assisted in quantifying the magnitude and spatial variability of the event with high precision. The third-party network could contribute to the understanding of the duration and spatial distribution of the storm, but underestimated the magnitude compared with the reference sensors.

## 1. Introduction

The global trend of urbanization is increasingly exposing people and assets to flood risks, which particularly affects the urban poor (Winsemius et al., 2018; Petersson et al., 2020; UN-Habitat, 2024). Flood mitigation and disaster preparedness measures require rainfall measurements on sub-hourly and sub-kilometre scale to be effective from the planning phase to post-event analysis (Guo, 2006; Marchi et al., 2009; Mailhot & Duchesne, 2010; Fuentes-



Andino et al., 2017; Pulkkinen et al., 2019; Imhoff et al., 2020). However, traditional monitoring
techniques generally have limited ability to accurately observe rainfall at this spatiotemporal
resolution. The most impactful rainfall events in urban areas are typically convective storms,
which can cause heavy rainfall over small areas and short durations with severe damage as
consequence (Kaiser et al., 2021; Mobini et al., 2021).
In Sweden, the Swedish Meteorological and Hydrological Institute (SMHI) operates around
600 rain gauges distributed over a land mass of 410,000 km$^2$. Of these, around 130 are
automatic stations recording accumulated rainfall depth every 15 minutes and the remaining
are manual stations reporting daily amounts. The station network is complemented with 12 C-
band Weather Radars (CWR) across the country with outputs every 5 minutes at 2 km spatial
resolution. While CWR generally is capable of producing good spatial representation of
precipitation, it has limitations caused by overshooting, beam blockage and clutter (Einfalt et
al., 2004; van de Beek et al., 2016). For highly localized convective events, the spatiotemporal
resolution of Sweden's official gauge network and radar composite is too low to capture
essential rainfall dynamics, such as spatial variability and peak intensity.
One option for national weather services to access high-resolution rainfall measurements is to
reach agreements with other professional entities like municipal water utilities and universities
who maintain their own monitoring networks, so-called "second-party data" (Garcia-Marti et al.,
2023). While these data might be trustworthy for operational use, their sampling resolution
may, just like official data, be insufficient on the "unresolved spatial scale" in which convective
storms occur (Lussana et al., 2023). In light of this, SMHI has recently gained interest in
additional external observations not operated by any official agency, sometimes referred to as
"third-party data". The new technologies are often enabled by digitalization and user-generated
content on the Internet, which lowers the barriers and costs associated with data acquisition.
While these data can provide higher resolution observations in space and time, they are often
subject to uncertainties and bias due to the lack of installation guidelines, maintenance
protocols and mechanisms to reinforce such standards. These promises and concerns have
sparked research efforts on applications and quality control of third-party data at SMHI and
many other European meteorological services (Hahn et al., 2022; Garcia-Marti et al., 2023).
This paper investigates the capability of second- and third-party rainfall sensors to observe a
highly localized convective storm that occurred on 18 August 2022 in Båstad, Sweden. The
second-party data come from sensors managed by local authorities in southwestern Sweden
and consists of a traditional rain gauge and an X-band Weather Radar (XWR). As for third-
party data, we study rainfall observations from a Commercial Microwave Link (CML) and a set
of Personal Weather Stations (PWS). CML and PWS are sometimes referred to as
"opportunistic sensors" (Fencl et al., 2024). Here, we will use the term "third-party data" for



consistency. First, the long-term (2021-2022) performance of the second-party rain gauge is
evaluated against the national weather stations to qualify as a trusted reference sensor for the
study. Then, an event analysis is performed by calculating evaluation metrics for each sensor
compared with the reference. Data from the radars and third-party sensors require pre-
processing and quality control to facilitate the analysis.
XWR are more low-cost compared with conventional C-band and S-band weather radars and
provide higher resolution imagery. They are on the other hand more affected by attenuation,
especially in widespread heavy rainfalls due to the accumulated attenuation throughout the
signal path (Lengfeld et al., 2016; Bobotová et al., 2022). XWR also have a shorter observation
range than conventional radars, typically 30-60 km (Thorndahl et al., 2017). CML are radio
links between base stations that connect the backbone of telecom networks to local
subnetworks (Chwala & Kunstmann, 2019). CML operate at frequencies where the
propagation of radio waves through the atmosphere is attenuated by rainfall. The transmitted
signal level (TSL) and received signal level (RSL) are collected by telecom companies for
network monitoring and maintenance purposes, so what is being considered as "noise" in
telecommunication can be used as a signal to estimate rainfall intensities for
hydrometeorological applications (Leijnse et al., 2007b).
A fundamental characteristic of CML observations is that they measure path-integrated rainfall,
generally assumed to represent the average rainfall intensity along the CML path. The validity
of this assumption has, however, been little investigated empirically. In this paper we study the
spatial variability of rainfall along a CML link by sampling XWR bins every 250 meters along
the CML reach, resulting in 20 XWR time series that are compared with the CML rainfall
estimates. This approach enables us to perform new investigations about bias in CML
observations due to variability of rainfall intensity along a CML path.
PWS are weather stations installed by occupants on their private property. Here, we consider
PWS that can be connected to online platforms to share observations openly in real time.
Recent years have seen a remarkable increase of PWS connected to the internet, presumably
due to the adoption of smart home technologies (Sovacool & Furszyfer Del Rio, 2020).
Contrarily to CML, PWS are designed to measure rainfall directly, but it can be assumed that
PWS data are subject to errors and bias linked to hardware, installation site and maintenance
(Boonstra, 2024). Various quality control protocols designed specifically for PWS have been
presented by literature (de Vos et al., 2019; Bárdossy et al., 2021; Lewis et al., 2021). However,
it has not been investigated how the algorithms perform when applied to localized extreme
rainfall. In this paper we apply an adjusted version of the PWS quality control protocol
suggested by de Vos et al. (2019) and compare the results with traditional evaluation metrics.



This paper addresses multiple gaps in the field of high-resolution rainfall monitoring by 1)
bench-marking second- and third-party sensors with an official monitoring network in the
context of convective rainfall, 2) cross-referencing radar observations with path-integrated
rainfall estimated by CML and 3) investigating the performance of a PWS quality control
protocol in the context of convective rainfall. The study is guided by the following research
questions:
• To what extent are second- and third-party sensors capable to observe convective
rainfall?
• What are the advantages and limitations when observing convective rainfall with
second- and third-party sensors, compared with a national monitoring network?
This paper is organized as follows. After this introductory section, section 2 presents the storm
event and area of interest that was selected for the case study. Section 3 describes the sensors
and data applied in the analysis. Section 4 presents evaluation metrics and methods applied
for the long-term and event analysis, as well as data processing.  Section 5 outlines the results
of the long-term and event analysis. Section 6 discusses the results, while section 7
summarizes the main findings of the study.

## 2. Case study

A convective rainfall event that hit the Bjäre Peninsula in southwestern Sweden, in the late
afternoon of 18 August 2022, was selected for the study. SMHI's forecast had indicated a small
likelihood of rainfall intensities above 35 mm/3h, which is the institute's threshold for rainfall
weather warnings. However, it was expected to hit further to the North, so no weather warning
was issued in the area at the time of the event. According to media reports, the rain was mixed
with hailstones of about 2 cm in diameter and caused flooding of around 60 buildings
(Gravlund, 2022; Bengtsson, 2023). A local water utility company (NSVA) operates a tipping
bucket rain gauge (hereafter 'municipal gauge') in the city of Båstad, which peaked at 216
mm/h and recorded 75.4 mm in 54 minutes. This corresponds to a return period of about 700
years, based on rainfall statistics developed for southwestern Sweden (Olsson et al., 2019).
The maximum depth recorded in 45 minutes was 71.2 mm, which breaks Sweden's official
record of 61.1 mm in 45 minutes at the *Daglösen* station in Värmland county on 5 July 2000.
The selected event was preceded by two dry days. The analysis focused on the urban area of
Båstad, a town with around 16,000 inhabitants located at the southern coast of the Laholm
Bay, covering approximately 9.4 km$^2$. Fig. 1 shows the locations of all sensors included in the
study.





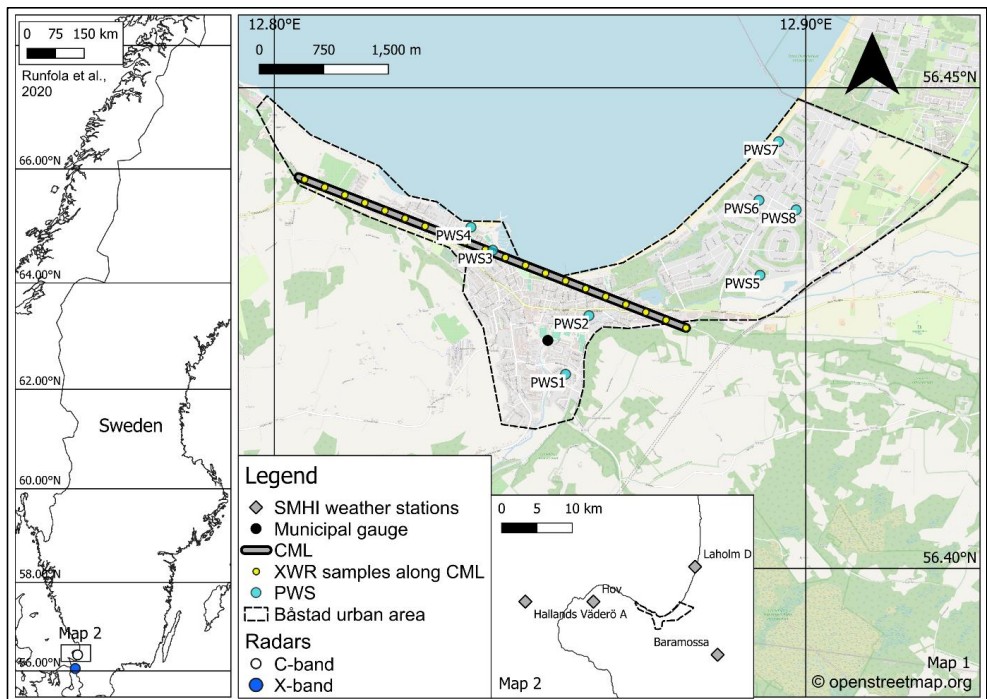

**Figure 1.** *Area of interest and locations of sensors. (Runfola et al., 2020).*

## 3. Data

Three levels of data were considered in the study – Sweden's national meteorological
monitoring network, a municipal gauge and XWR operated by local and regional agencies
(second-party network) and CML and PWS (third-party network). More details on the data
sets are provided below.

### 3.1 National monitoring network

The national weather monitoring network operated by SMHI consists of a combination of
manual and automatic weather stations and CWR. The *Hov*, *Laholm D* and *Baramossa*
weather stations, located 9-10 km away from Båstad (Fig. 1), report daily accumulated rainfall
at 06:00 UTC+2, manually observed by certified observers. The automatic rain gauge station
of weighing type on the island *Hallands Väderö*, situated 15 km west of Båstad, report 15-
minute accumulations. As these data have passed quality assurance protocols at SMHI, we
consider them the most trustworthy source to use for benchmarking in the study. Precipitation
data from the stations for the year 2022 was downloaded from SMHI's open data archive
(SMHI, 2025a).





In addition, we studied a gauge-adjusted Plan Position Indicator (PPI) radar composite based
on the lowest elevation scan (0.5°) from all radars operated by SMHI. While the radars can
operate in dual-polarization mode, this product is based on the horizontal polarization. Radar
reflectivity $Z$ [mm$^6$/m$^3$] can expressed as integrals over the Drop Size Distribution (DSD) in the
pulse volume, here $N(D)$ [mm/m$^3$].

$$Z = \int_0^\infty D^6(D)N(D)dD \qquad (1)$$

where $D$ [mm] is the spherical drop diameter. The closest radar is situated 6 km south of
Båstad, (Fig. 1). Since this radar was operational during the selected event, the studied
composite is based on data from only this radar during the period of interest. The composite is
available in 5 minutes resolution at a spatial resolution of 2x2 km. The compositing of radar
data at SMHI is done with the *BALTRAD* software (*BALTRAD*, n.d.) The composite was
downloaded from SMHI's open radar archive where it is distributed as radar reflectivity data in
GeoTIFF-format (SMHI, 2025b).
Figure 2 shows the elevation profile and radar beam profile between the CWR location and the
location of the municipal gauge in Båstad. The low elevation angle and short distance to the
area of interest indicate that the observations are made at approx. 200-300 m above sea level,
eliminating the risk of overshooting, as convective precipitation in the summer months in
Sweden typically originate from much higher altitudes. However, SMHI has experienced partial
beam blockage caused by vegetation within 1 km north of the radar location (SMHI, 2020),
which is known to affect observations over Båstad.

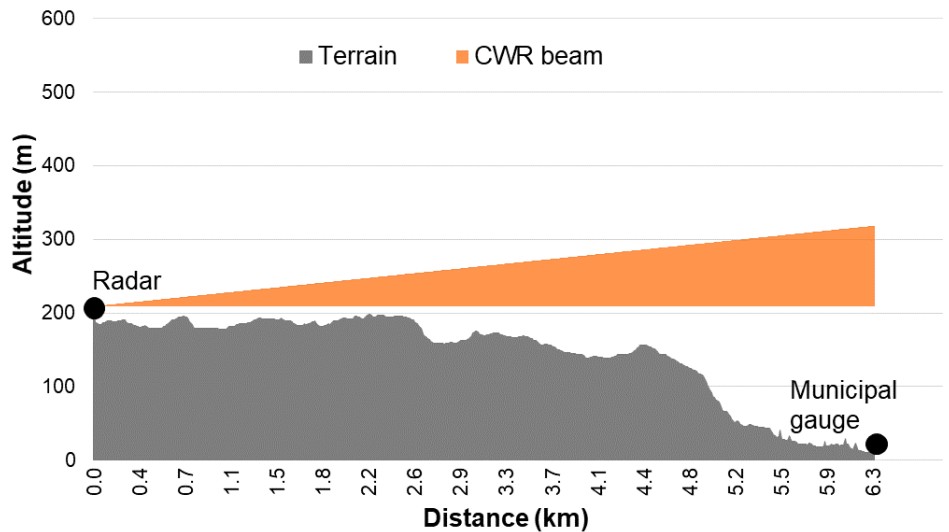


*Figure 2. Elevation profile and beam profile between CWR radar location and municipal gauge*





## 3.2 Second-party monitoring network


We consider two second-party sensors operated by local and regional authorities; a municipal
gauge in Båstad managed by the local water utility company NSVA, and a compact FURUNO
dual-polarization XWR operated by NSVA on behalf of Lund University. The municipal gauge
is a Casella tipping bucket which records a tip each time the bucket volume (0.2 mm) is filled
on 1-second resolution. Time series with 1-minute resolution from the municipal rain gauge for
the years 2021-2022 were received upon request from NSVA.
The XWR is located in Helsingborg, 40 km south of Båstad (Fig. 1). The spatial resolution of
the data is 0.5 degrees of azimuth and 75 m of slant range at 1-minute time intervals. XWR
data for the day of the event was acquired from VeVa (*Weather Radar in the Water Sector*)
(VeVa, n.d.), a collaboration between water utility companies in south Sweden and Denmark
that distributes XWR data to its partners in according to the EUMETNET Opera Data
Information Model (Michelson et al., 2014). The pre-calculated rainfall rate (mm/h) from the
lowest scan (elevation angle of 1°) was used, which integrates dual-polarization variables as
a method for attenuation correction as described in detail in Hosseini et al. (2020).
Figure 3 shows that the XWR's half-beam vertical profile has a larger sampling volume and
steeper elevation angle than CWR (Fig. 2) at the area of interest. As the profile extends 300-
1200 meters above sea level over Båstad, this, just like for CWR, suggests a very small risk
of signal contamination due to beam blockage and overshooting.

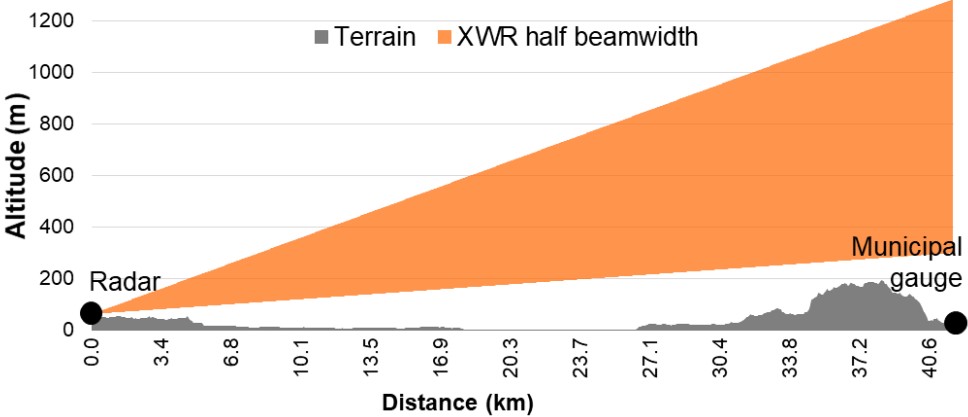


*Figure 3. Elevation profile and half beam profile between XWR radar location and municipal gauge.*

## 3.3 Third-party monitoring network


CML data were received as TSL and RSL at 10 seconds resolution upon request from the
telecom companies Ericsson AB and Tre. The data covered all base stations on the Bjäre
Peninsula for the days 18-19 August 2022. Each antenna works as both transmitting and



receiving terminal, meaning that each link has bidirectional transmission and provides at
least two radio signals. Here, we use the term 'sub-link' to refer to a single radio signal. The
location of the only CML in the area of interest is shown in Fig. 1. The link is approximately
4.8 km long. Further details on CML processing is provided in section 4.4.3.
The selected PWS type in this study, NetAtmo, is an unheated plastic tipping bucket rain gauge
that reports the number of tips through a wireless connection to the accompanying indoor
module (de Vos et al., 2019). The indoor module broadcasts the observations to Netatmo's
online platform at approximately 5-minute intervals. The default tipping bucket volume is 0.101
mm, or another volume specified by the station owner using the product's calibration feature.
PWS time series for the study were received from NetAtmo.

## 4. Methods

The analyses covered two stages – a long-term analysis and an event analysis. This section
first presents the evaluation metrics applied to assess the performance of the sensors in the
study, followed by descriptions of the methods applied in the long-term analysis and event
analysis. Then, the data processing performed on radar and third-party data is described.

### 4.1 Evaluation metrics

Three evaluation metrics were used to assess the performance of each sensor: Spearman
rank correlation ($r_s$), Root Mean Squared Error (RMSE), and Percent Bias (PBIAS). The
absolute difference in total rainfall depth during the event was also calculated.
The Spearman correlation is a non-parametric test that measures the strength of a monotonic
relationship between two variables:

$$r_s = 1 - \frac{6 \sum d_i^2}{n(n^2 - 1)} \tag{2}$$

where $d_i$ is the difference between ranks for each pair of values and $n$ is the number of
observations. The closer to -1 or 1, the better the negative or positive monotonic relationship.
If very low correlations (close to 0) were found, time lags were applied iteratively by shifting
one of the time series a few minutes back or ahead in time to see if this could increase the
correlation. This can be expected when cross-referencing observations of convective storms,
as the highly intermittent nature of rainfall can cause low correlations even if the total rainfall
depth is similar at two nearby locations. As the Spearman correlation does not address the
magnitude of error, it can be complemented with RMSE (Hyndman & Koehler, 2006):





$$RMSE = \sqrt{\frac{1}{n}\sum_{i=1}^{n}(O_i - T_i)^2} \qquad (3)$$

where $O_i$ is the reference rainfall and $T_i$ is the evaluated data. Lower RMSE indicates a better
model performance. Finally, PBIAS quantifies the average bias, where a positive or negative
value suggests an underestimation or overestimation of rainfall depth, respectively (Gupta et
al., 1999):

$$PBIAS = 100 \times \frac{\sum_{i=1}^{n}(O_i - T_i)}{\sum_{i=1}^{n} O_i} \qquad (4)$$

## 237 4.2 Long-term analysis

As the magnitude of the selected event was not captured by the national network (see Results)
it was necessary to establish another reliable reference for the event analysis. Consequently,
the long-term (2021-2022) performance of the municipal gauge was evaluated against the
national weather stations by applying the metrics presented in section 4.1. The gauge was
cross-referenced with the manual stations *Hov, Laholm D* and *Baramossa* operated by SMHI,
all situated 9.3-9.7 km away (Fig. 1). The station *Hallands Väderö A* was excluded from the
comparison as it is located on an island 15 km west of Båstad. The tips recorded by the
municipal gauge was resampled to daily accumulations between 06:00-05:59 UTC+2, as this
is the sampling frequency of the reference (manual) stations.

## 247 4.3 Event analysis

The event analysis was performed by plotting rainfall rate *P* and total depth *D* observed by the
second- and third-party sensors and calculating the evaluation metrics. The return period of
the event was calculated based on SMHI's climate statistics for southwestern Sweden (Olsson
et al., 2019). Based on performance, it was decided to exclude CWR as reference sensor
(section 5.2.2). The municipal gauge served as reference for the CWR and XWR observations.
After concluding that XWR recorded similar rainfall depth as the municipal rain gauge during
the event, XWR was used as reference for the third-party data to better account for the spatial
variability of the event. XWR data were sampled every 250 m along the reach of the CML to
investigate the variability of rainfall intensity along the link (section 4.4.4). For PWS, a quality
assurance protocol was applied (section 4.4.5).
The temporal range of the studied event differed between the sensors as the start and end of
the rainfall occurred at different times in the observed time series. The event start was defined
as the first timestep when it had been raining more than 0.1 mm/h for at least 5 minutes at the
reference sensor and event stop when it had been raining less than 0.1 mm/h for at least 5



minutes. The calculation of evaluation metrics, return periods and accumulated depths were
calculated for this temporal range only, to reduce the number of timesteps with zero rainfall in
the calculations, which could distort the relationships.

### 4.4  Data processing

The radars and third-party data required different levels of pre-processing and quality control
which are outlined in the following sections.

#### 4.4.1 C-band weather radar

Radar reflectivity $Z$ (Eq.1) is generally expressed logarithmically as *dBZ*.

$$dBZ = 10 \times \log_{10}(Z) \tag{5}$$

The CWR composite retrieved from SMHI's radar archive is distributed as pseudo-dBZ $E$
(integer 0-255), which was translated to *dBZ* by applying gain and offset:

$$dBZ = E \times G + offset \tag{6}$$

where $G$ is the gain. Here, $G$ = 0.4 and *offset* = -30 (Michelson et al., 2014). The rain rate $P_{CWR}$
(mm/h) can be found from the reflectivity following an inverted power law relationship:

$$P_{CWR} = \left(\frac{Z}{a}\right)^{\frac{1}{b}} \tag{7}$$

We applied the parameters suggested by Marshall & Palmer (1948), $a$=200 and $b$=1.6. The
actual values of $a$ and $b$ can vary greatly depending on the actual DSD, which may be different
within and from event to event (Battan, 1981).
The resulting geoTIFF-files were sampled at the location of the municipal gauge and the eight
PWS. In this way, time series at 5-minute resolution from the CWR composite were created.
The accumulated rainfall $D_{CWR}$ (mm) was then calculated per point of interest and for the CWR
grid. Evaluation metrics were calculated for the duration of the event as recorded by the
municipal gauge.

#### 4.4.2 X-band weather radar

For XWR, the manufacturer's built-in precalculated rainfall rate $P_{XWR}$ (mm/h) on 1-minute
resolution was used for the study. The underlying equations for calculating the rainfall rate is
generally similar to CWR as described in the previous section. However, a main difference is
that XWR uses different coefficients and corrected dBZ based on the dual-polarization
variables, which has been shown to be useful for summer precipitation estimations in
Sweden (Hosseini et al., 2020, 2023). It is also noted that the XWR data were available in
polar bins, that is, range gates at a given elevation and azimuthal angle, in contrary to the



regular cartesian grids for the utilized CWR data. Thus, time series were extracted from the
XWR bins closest to the projected locations of interest, accounting for elevation, range
difference and azimuth difference. The sampled points included the municipal gauge, the
eight PWS, and 20 points along the CML path as described in section 4.4.4.
A few missing values were found in the XWR time series, which occurred during the most
intense part of the storm. These were filled with linear interpolation. The volumetric data was
gridded into a cartesian grid 500 meters using the *wradlib* Python package (Mühlbauer &
Heistermann, 2024). The accumulated rainfall $D_{XWR}$ (mm) was then calculated per point of
interest and for the XWR grid. Evaluation metrics were calculated for the duration of the
event as recorded by the municipal gauge.

### 4.4.3 Commercial Microwave Links

When estimating rainfall intensity from CML data, the first step is to identify a link-specific
threshold for classification of wet and dry timesteps. The challenge is to detect small rainfall
volumes (true wet periods) without including too many dry periods with strong attenuation
from other causes, such as changes in water vapour content or air temperature (false wet
periods). Several approaches have been suggested in literature (Rayitsfeld et al., 2012;
Wang et al., 2012; Cherkassky et al., 2014; Overeem et al., 2016). Schleiss & Berne (2010)
proposed a simple classification method that considers the rolling standard deviation of the
attenuation, assuming that the variability is small during dry periods and large during wet
periods. The time step is classified as dry if the variability falls below a defined threshold
value, which must be calibrated with secondary observations nearby the link. More recently,
machine learning approaches has shown strong potential to effectively classify wet and dry
timesteps in CML data (Habi & Messer, 2018; Polz et al., 2020; Øydvin et al., 2024).
The second step is to define a 'baseline level', that is, RSL during dry weather. This is used as
the reference level for the rain attenuation calculation and is typically based on the signal
attenuation during dry time steps preceding a wet period (Andersson et al., 2022). In addition,
the signal is often corrected for additional attenuation caused by water on the cover of the
antenna, so-called 'wet antenna attenuation' (e.g., Leijnse et al., 2007a, 2008; Graf et al.,
2020). Finally, the corrected attenuation is converted into rain rate using an inverted power law
relationship.
Received TSL and RSL were converted into rainfall rate using the MEMO (Microwave-based
Environmental Monitoring) method developed by SMHI  (SMHI, 2025c). This method follows
the general steps applied by most CML algorithms as desribed above. However, the method
does not explicitly correct for wet antenna attenuation, but instead applies a bias correction





factor $CF_A$ based on link length to the derived rain rate $P_{raw}$ (mm/h) that compensates for the wet antenna effect:

$$P_{CML} = P_{raw} - (A_{nl} * CF_A) \tag{8}$$

Here, $P_{raw}$ is the uncorrected rainfall intensity and $A_{nl}$ is the net attenuation. Details on the processing steps of the MEMO methodology are outlined in Appendix A.

The CML included in this study consists of two sub-links. These recorded very similar values, with a difference in total rainfall depth of 3 mm for the whole event. Thus, the mean rain rate $P_{CML}$ and mean depth $\bar{D}_{CML}$ per timestep of the two sub-links were used in the analysis. Evaluation metrics were calculated for the duration of the event as recorded by the mean of 20 XWR bins sampled along the CML reach, see further details in the next section.

### 4.4.4 XWR and CML analysis along CML path

To investigate how CML observations of extreme rainfall are impacted by spatial variability along the link, XWR bins were sampled each 250 m along the reach of the CML, resulting in 20 XWR time series on 1-minute resolution (Fig 1.) The mean of the 20 XWR bins $\bar{P}_{XWR}$ was used as reference for CML. The 10th and 90th percentiles were calculated to explore the range of $P_{XWR}$ and $D_{XWR}$ along the CML path. The behaviour of the XWR data along the CML reach during the intense part of the storm was inspected visually. Hypothesizing that the difference in XWR and CML observations is related to the XWR variability along the link, an ordinary least squares analysis was performed on the difference $\bar{P}_{XWR}$ and $\bar{P}_{CML}$, with the XWR standard deviation as independent variable.

### 4.4.5 Personal Weather Stations

PWS data were received as *.csv*-files and processed into *netCDF*-files following the standards proposed by Fencl et al., (2024). PWS without a rainfall sensor, and PWS that were offline during the storm event, were excluded from the analysis. This resulted in a total of eight PWS located within the Båstad urban area (Fig. 1). For each PWS, the evaluation metrics were calculated compared with XWR time series sampled at the same location. The PWS timeseries were processed with a quality control package as described below.

Research has shown that the quality of rainfall data from PWS can be improved significantly by applying quality control and bias correction. The algorithms suggested in literature (see, e.g, Mandement & Caumont (2020), Lewis et al. (2021), Bárdossy et al. (2021)) typically utilize the high observation density of PWS by comparing rainfall time series with the performance of neighbouring stations, referred to as '*buddy checks'* by Båserud et al. (2020). De Vos et al. (2019) developed a quality control protocol for PWS rainfall data in the R programming language, *PWSQC*. The method does not rely on a primary monitoring network, but flags suspicious measurements based on the observations from nearby stations. The method has



been applied in gauge-adjustment of radar by Nielsen et al. (2024) and Overeem et al. (2024)
and has recently been converted to a Python package, *pypwsqc*, that was applied for the study
(Chwala et al., n.d.).
Event time series from the eight PWS were processed with *pypwsqc*. The algorithm applies
three filters utilizing neighbour checks – the Faulty Zeroes filter, High Influx filter, and Station
Outlier filter – to assess the quality of each time step in rainfall time series by comparing with
the records of neighbouring PWS within a user-defined radius (refer to de Vos et al., 2019, for
details). The Faulty Zeroes filter flags timesteps when the evaluated station records zero
rainfall for at least $n_{int}$ time intervals, while the median of the surrounding rainfall observations
is larger than zero. The High Influx filter identifies unrealistically high rainfall mounts based on
a comparison with the median rainfall of the neighbouring stations. The Station Outlier filter
flags a station as an "outlier" if the Pearson correlation with the median rainfall of neighbours
in a selected evaluation period falls below a set threshold.
To improve the performance of the neighbouring checks, data from all PWS within a 10 km
radius around Båstad were considered, which resulted in a total of 58 stations. However, only
results of the 8 PWS within the area of interest were evaluated during the event. To get a better
understanding of the long-term performance of each PWS, the quality control was also applied
for the full year of 2022. The parameters were set to the same values as in the original
publication (de Vos et al., 2019), except $m_{match}$ and $m_{int}$. These parameters control how many
overlapping wet time steps between the evaluated PWS and its neighbours that are needed
within a defined evaluation period to reliably apply the Station Outlier filter. The numbers
proposed by de Vos et al. (2019) were found to be too strict for the PWS dataset in this study,
as the Station Outlier filter could not be applied for very long periods. Instead, the parameters
were adjusted to require less wet time steps during a longer evaluation period.  Table 1 shows
the parameter settings used in the quality control.
***Table 1.*** *Parameter settings for PWS quality control. See de Vos et al. (2019) for description of each*
*parameter.*

| Parameter | Value |
|---|---|
| d (m) | 10,000 |
| $n_{stat}$ | 5 |
| $n_{int}$ | 6 |
| $\Phi_A$ (mm) | 0.4 |
| $\Phi_B$ (mm) | 10 |
| $m_{int}$ | 8064 |
| $m_{match}$ | 100 |





| γ | 0.15 |
|---|---|


## 5. Results

### 5.1 Long-term analysis

Figure 4 shows daily accumulations from the *Hov*, *Laholm D* and *Baramossa* weather stations
and the municipal gauge for the years 2021 and 2022. The plot shows that the municipal gauge
recorded significantly less rainfall than *Baramossa* but followed *Hov* and *Laholm D* reasonably
well. This may be because the rain gauges in Båstad, Hov and Laholm are all located close to
the coast, whereas the records in Baramossa represent an inland climate.

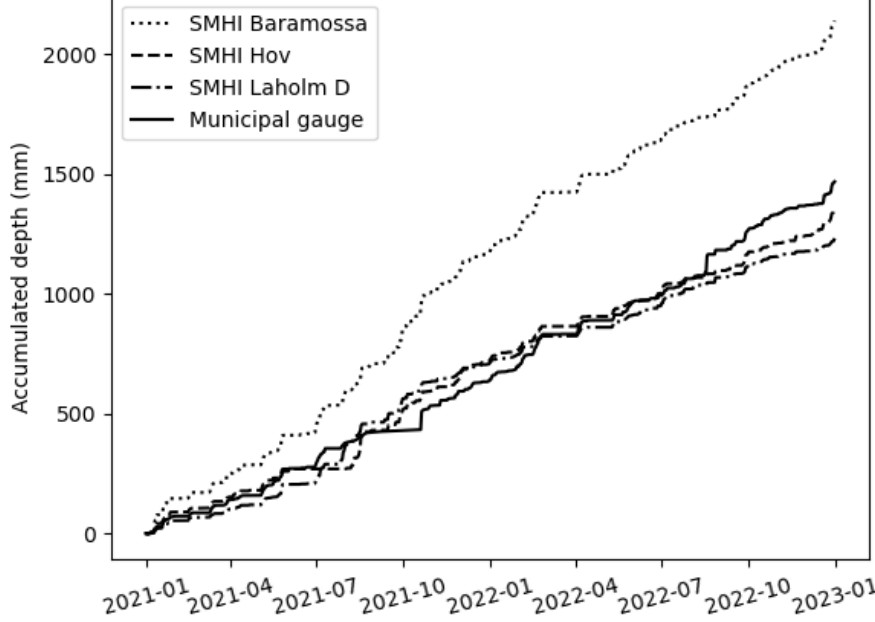


***Figure 4.*** *Accumulated depth 2021-2022 for municipal gauge and SMHI rain gauges located 9.3-9.7 km*
*away.*
Table 2 shows the evaluation metrics of the municipal gauge, benchmarked with the three
reference stations. The PBIAS over 2 years was only -8% compared with *Hov* weather
station, which is considered low as the stations are situated 9.7 km apart. Based on these
results, the municipal gauge was accepted as a trusted reference for the event analysis.



*Table 2. Cross-validation of the municipal gauge with three reference stations, 2021-2022.*

| Reference station (SMHI) | Distance to municipal gauge (km) | $r_s$ | RMSE (mm/day) | Absolute accumulated difference (mm) | PBIAS (%) |
|---|---|---|---|---|---|
| Baramossa | 9.4 | 0.55 | 6.1 | 674 | 31% |
| Hov | 9.7 | 0.46 | 5.87 | 118 | -8% |
| Laholm | 9.3 | 0.52 | 5.23 | 233 | -19% |


## 5.2 Event analysis

### 5.2.1 Event duration

Table 3 summarizes the event duration observed by each sensor. National weather stations
were excluded from the analysis, either because they record daily precipitation, or recorded
very small total depth (section 5.2.2). The municipal gauge recorded rainfall for 54 minutes,
which is among the shortest durations with only PWS 4 observing rain for a shorter period
(50 min). Notably, XWR recorded rain for 109 minutes at the location of the municipal gauge.
This follows the general pattern that XWR recorded rain for a longer period than the
corresponding gauge. The difference was generally around 30 minutes, possibly due to the
higher sensitivity of XWR to light drizzles, either never reaching the ground or slowly
accumulating in the tipping bucket before the first tip was recorded at the weather station.
Comparing XWR with CML, there was only 4 minutes difference in observed event start.
The PWS are concentrated in two clusters. PWS 1-4 are located in the western and central
part of Båstad together with the municipal gauge, and PWS 5-8 in the north-eastern part (Fig
1). In Table 3 it can be seen that ground observations in the mid-western part of Båstad
started recording rain between 16:55 and 17:15, and the north-eastern part between 17:15
and 17:25, which suggests a gradual motion of the storm from west/south-west to the north-
east. A similar tendency is seen in the XWR data, but with approximately 30 minutes time
lag.
*Table 3. Event duration observed by each sensor.*

| Sensor | Type | Event start (UTC+2) | Event end (UTC+2) | Duration (min) |
|---|---|---|---|---|
| Municipal gauge | reference | 17:15 | 18:09 | 54 |
| XWR at municipal gauge | test | 16:45 | 18:34 | 109 |
| CWR at municipal gauge | test | 16:45 | 17:55 | 70 |
| XWR mean along CML | reference | 16:38 | 18:36 | 118 |
| CML mean | test | 16:34 | 18:43 | 129 |
| XWR at PWS 1 | reference | 16:40 | 18:30 | 110 |
| PWS 1 | test | 17:05 | 18:30 | 85 |



| XWR at PWS 2 | reference | 16:45 | 18:30 | 105 |
| PWS 2 | test | 17:10 | 18:30 | 80 |
| XWR at PWS 3 | reference | 16:40 | 18:30 | 110 |
| PWS 3 | test | 16:55 | 18:10 | 75 |
| XWR at PWS 4 | reference | 16:40 | 18:10 | 90 |
| PWS 4 | test | 16:55 | 17:45 | 50 |
| XWR at PWS 5 | reference | 16:45 | 18:35 | 110 |
| PWS 5 | test | 17:25 | 20:00 | 155 |
| XWR at PWS 6 | reference | 16:45 | 18:35 | 110 |
| PWS 6 | test | 17:15 | 18:35 | 80 |
| XWR at PWS 7 | reference | 16:50 | 18:35 | 105 |
| PWS 7 | test | 17:25 | 18:35 | 70 |
| XWR at PWS 8 | reference | 16:45 | 18:35 | 110 |
| PWS 8 | test | 17:20 | 18:30 | 70 |


### 5.2.2 National monitoring network

The total rainfall depth observed by the national monitoring network is shown in Fig. 5. The
CWR grid size is 2x2 km. The weather station *Hallands Väderö A*, situated 15 km west of
Båstad, record accumulated values every 15 minutes but only observed a total volume of 0.4
mm on the day of the event. The other stations report daily accumulations between 06:00-
05:59 UTC+2, amounting to a maximum depth of 14.2 mm at *Hov*. All observations from
SMHI's gauges corresponded to a return period of less than 1 year (Olsson et al., 2019). The
heaviest rainfall observed by CWR was concentrated to the south of Båstad urban area, with
a maximum total depth of 65 mm, which corresponds to a return period of around 400 years
for a duration of 60 minutes (Olsson et al., 2019). The maximum recorded depth in the area
of interest was 25 mm (to the south-east), which corresponds to a return period of 11 years
for a duration of 60 minutes.



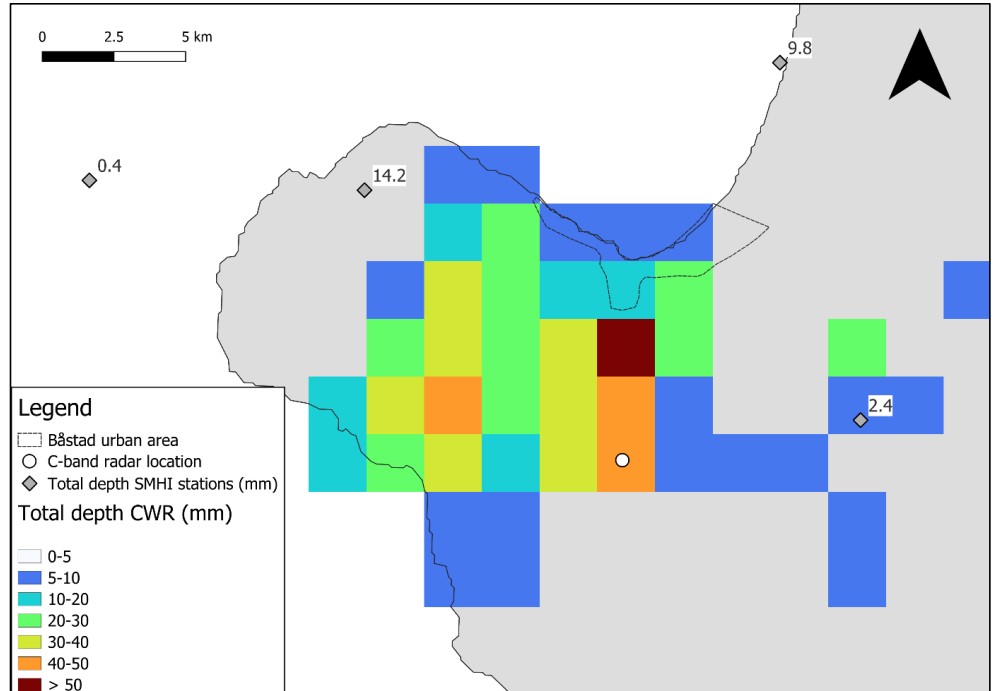

**Figure 5.** *Total accumulated depth of the event recorded by the national monitoring network.*

Figure 6 shows the rainfall event observed by the municipal gauge, compared with CWR sampled at the same location. CWR underestimated the total depth with 57 mm when comparing with the gauge, which suggest that CWR could not quantify the magnitude of the event accurately. The CWR started to observe rain 30 minutes before the rain gauge. Different time lags were applied to the time series by iteration, and it was found that $r_s$ could be raised from 0.07 to 0.77 when adding a lag of 25 minutes to the CWR data.



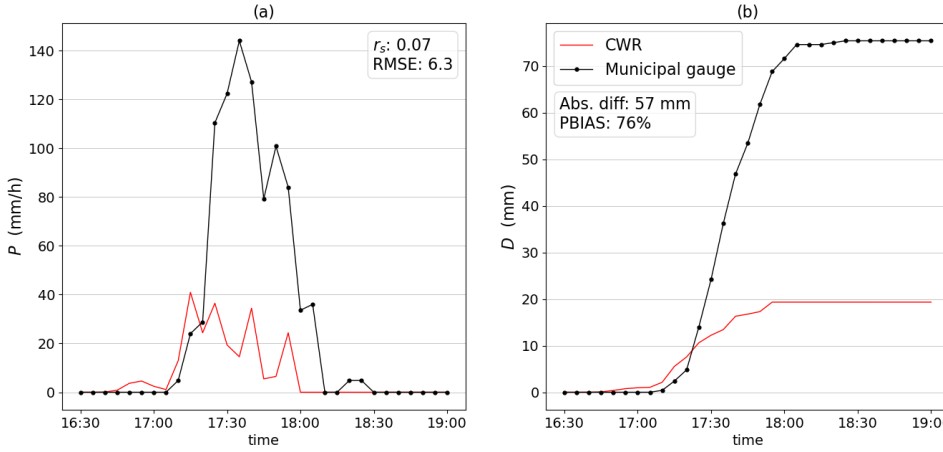

***Figure 6.*** *a) Rainfall intensity P (mm/h), Spearman rank coefficient $r_s$ (-) and RMSE (mm/5min). b) Total depth D (mm), absolute difference in total depth and PBIAS.*

### 5.2.3 Second-party monitoring network

Figure 7 shows the total accumulated depth observed by the second-party network. The municipal gauge observed rainfall for 54 minutes, between 17:15-18:09, which is here approximated as 60 minutes. This corresponds to a return period of around 700 years for a total depth of 75.4 mm (Olsson et al., 2019). The location of the heaviest rainfall was different when comparing gridded XWR data to the CWR grid. The XWR data indicated two hotspots; one with total depths above 50 mm in large parts of the Båstad urban area, and one with a maximum depth of 61 mm 5 km south-west of the city. However, the total depth of XWR sampled at the location of the municipal gauge based on the closest XWR bin was 78.4 mm, corresponding to a return period of around 800 years for 60 minutes duration. The total depth in the gridded XWR data was lower than the point observation because of the averaging that occurs when interpolating the grid. XWR observations above 5 mm occurred over a much larger area compared with the CWR, especially to the north-east. This suggests that the CWR might have been affected by beam blockage during the event.





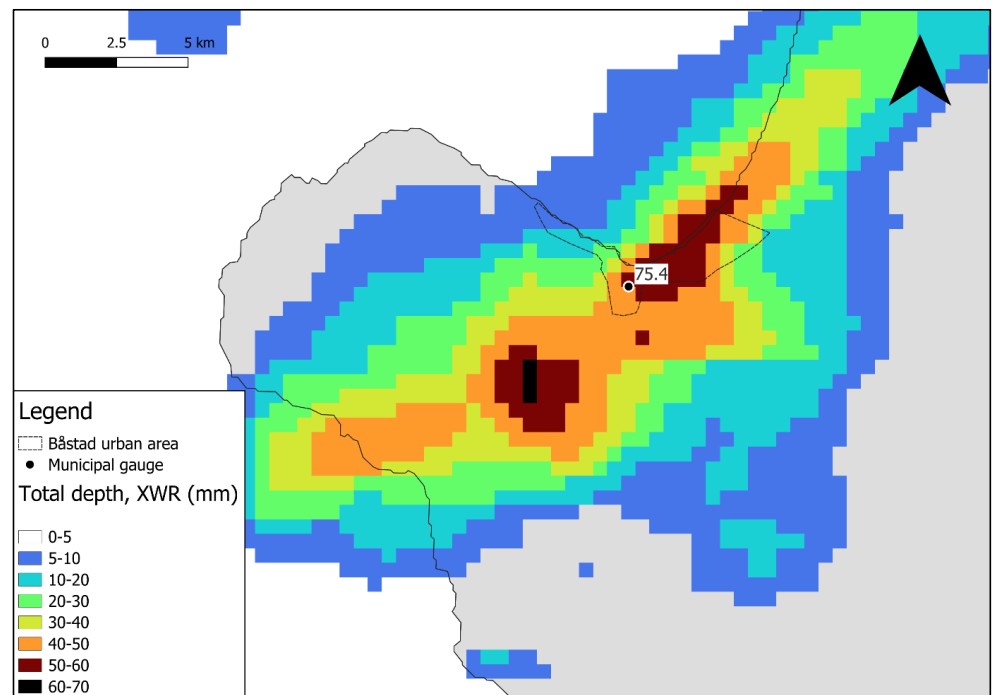

**Figure 7.** *Total accumulated depth of the event recorded by the regional monitoring network.*

Figure 8 shows the rainfall event observed by the municipal gauge, compared with XWR sampled at the same location. Similar to CWR, the XWR started to record rain almost 30 minutes before the rain gauge. The correlation $r_s$ could be raised from 0.18 to 0.39 when subtracting a lag of 10 minutes from the XWR data. Even if the correlation was low with the reference, XWR observed a similar total depth with only 3 mm overestimation. In Fig. 8a, it seems that XWR underestimated peak rainfall intensity and overestimated low rainfall intensity. This might be related to signal attenuation during heavy rain and higher sensitivity of XWR to drizzles or observations of melting particles during light rain.



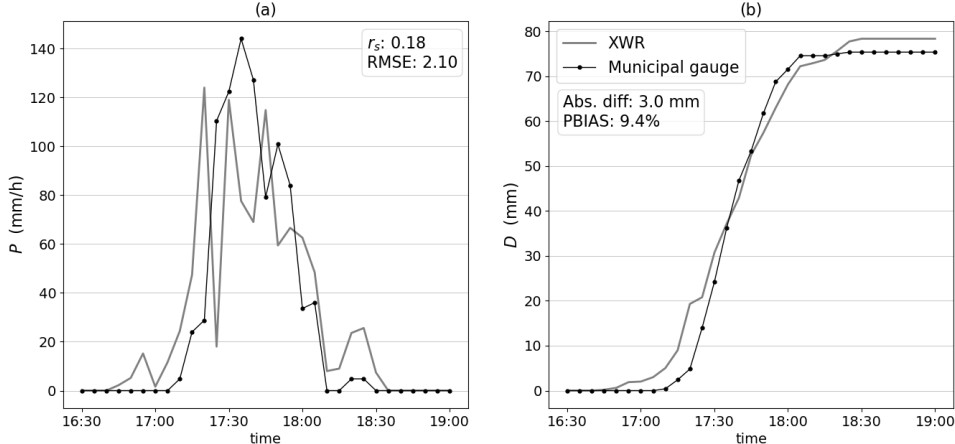

470

***Figure 8.*** *a) Rainfall intensity P (mm/h), Spearman rank coefficient $r_s$ (-) and RMSE (mm/5min). b) Total*

*depth D (mm), absolute difference in total depth and PBIAS.*

### 5.2.4 XWR and CML analysis along CML path

Figure 9 shows the rainfall intensity $\bar{P}_{CML}$ and depth $\bar{D}_{CML}$ expressed as the mean of the two

CML sub-links and the $10^{th}$-$90^{th}$ percentiles of the XWR bins sampled along each 250 m

(amounting to 20 sample time series) along the CML path. The mean intensity of the XWR

samples $\bar{P}_{XWR}$ is highlighted in grey and was used as reference for the CML. XWR on average

started to observe rainfall at 16:38 along the link path, and CML at 16:43. $\bar{P}_{CML}$ reached a

'plateau' at 83 mm/h and stayed almost constant at this level for 31 minutes between 17:27-

17:58. This effect is caused by complete loss of radio signal between the CML base stations,

which is induced by the heavy rainfall, as described by Blettner et al. (2023) and Polz et al.

(2023). Despite the plateau, the metrics are remarkably good with high correlation (0.9) and 8

mm total underestimation compared with the reference. The large spread of 10-$90^{th}$ percentiles

obtained from the 20 XWR observations suggests a large spatial variability of rainfall along the

link.



486

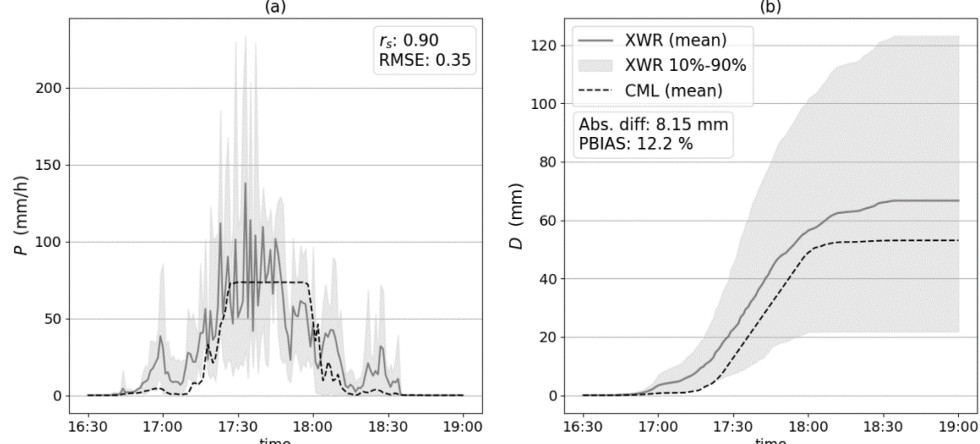

487

**Figure 9.** *a) Rainfall intensity P (mm/h) of CML (mean) and XWR (mean and 10<sup>th</sup> to 90<sup>th</sup> percentile) along CML path. Spearman rank coefficient $r_s$ (-) and RMSE (mm/1min). b) Total depth D (mm) of CML (mean) and XWR (mean and 10<sup>th</sup> to 90<sup>th</sup> percentile) along CML path. Absolute difference in total depth and PBIAS.*

By inspecting radar fields, it was observed that the storm propagated almost perpendicularly over the CML link, which is favourable for a detailed comparison between the XWR and CML observations over the link path. The CML plateau period was considered unsuitable for comparison, so the following analysis focused on the periods right before and after the signal loss, from 16:38-17:26 and 17:59-18:36, in total 85 minutes.

Figure 10 shows the rainfall intensity distribution along the CML as observed by XWR for five minutes before and after the plateau. The first bin to the left in the plots was sampled at the western end of the CML, approximately 3.4 km away from the municipal gauge, and the last bin to the right was sampled at the eastern end, 1.6 km away from the gauge (see Fig 1.). The XWR sampling points closest to the rain gauge (bin 14 and 15, counting from the left) are at approximately 700 meters' distance from the gauge. $P_{CML}$ and $P_{municipal}$ are also shown for each time step. The XWR spatial distribution was sometimes rather smooth, with gradual increase and decrease along the link (e.g., Fig. 10b), but sometimes more intermittent, with large differences between adjacent XWR samples (e.g., Fig 10g). In the pre-plateau period (Fig. 10a-10e) $\bar{P}_{CML} < \bar{P}_{XWR}$ consistently, whereas in the post-plateau period the relation was generally the opposite (Fig. 10f-10j).





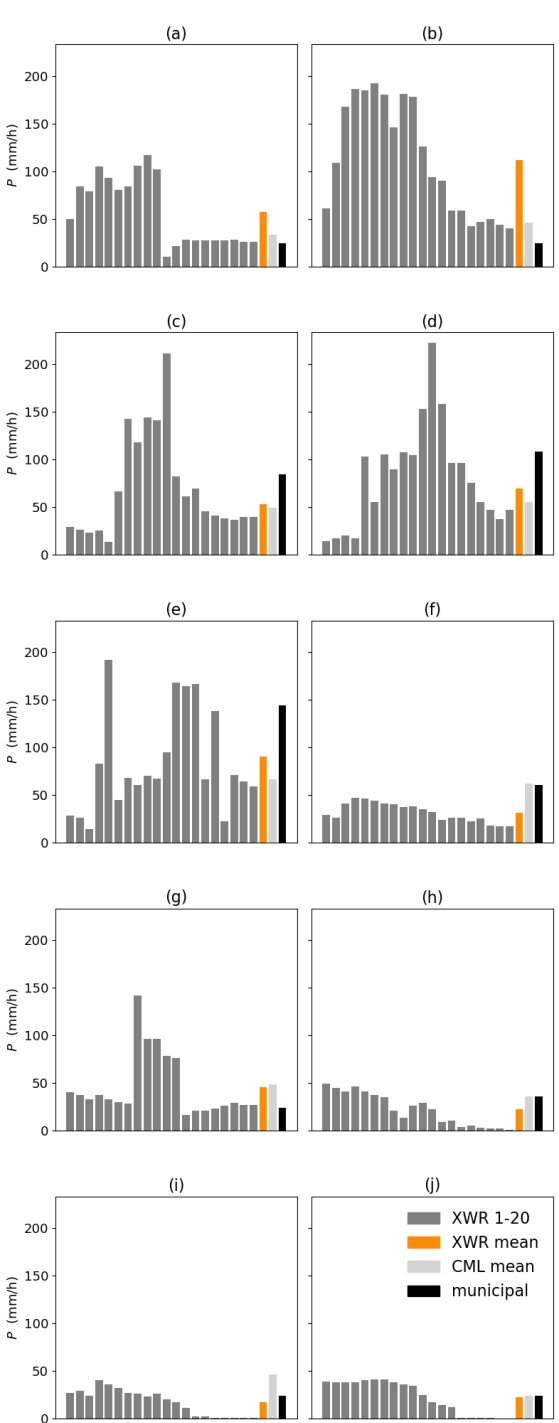





**Figure 10.** *(a)-(e) Rainfall intensity 5 minutes before CML signal loss (17:22-17:26). (f)-(j)*
*Rainfall intensity 5 minutes after signal loss (17:59-18:03). Twenty radar bins sampled along*
*CML path every 250 meters (XWR 1-20), XWR mean, CML mean and municipal gauge.*
The relationship between $\bar{P}_{XWR}$ and $\bar{P}_{CML}$ is shown for all observations in the pre- and post-
plateau periods (in total 85 data points) in Fig. 11a. $\bar{P}_{CML}$ was generally lower, and especially
when $\bar{P}_{XWR} < 20$ mm/h, then $\bar{P}_{CML}$ was consistently very low.  This suggests that XWR is more
sensitive to light rain than CML, as was observed when comparing with the municipal gauge
(section 5.2.3). Hypothesizing that the difference between $P_{XWR}$ and $P_{CML}$ was related to the
XWR variability over the link, Fig. 11b shows the difference as a function of the X-band
standard deviation $\sigma_{XWR}$. Despite a substantial scatter, a reasonably linear trend is suggested
(R²=0.31) with $\bar{P}_{CML}$ gradually underestimating more as the standard deviation increases.

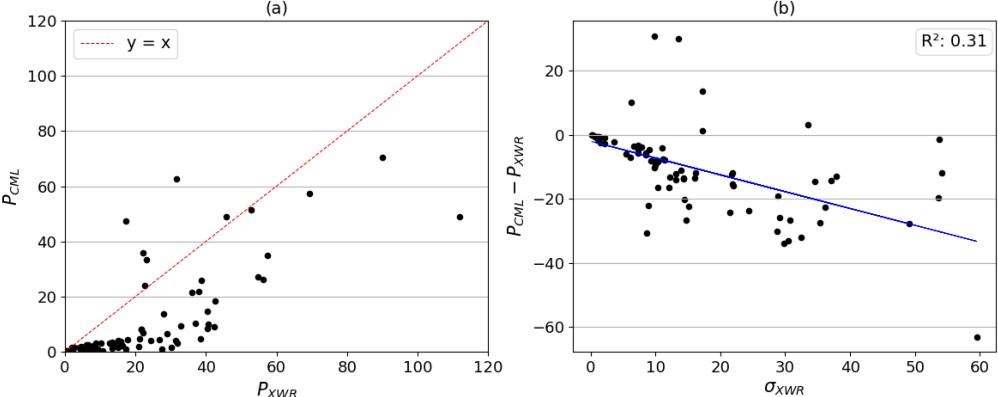

**Figure 11.** *a) Mean rainfall intensity P (mm/h) along the CML link as estimated by CML and XWR*
*observations for 85 timesteps. b) Difference between CML and XWR mean intensity values as a function*
*of XWR standard deviation $\sigma_{XWR}$ along the CML link.*
### 5.2.5 Personal Weather Stations
This section starts with the results of the PWS quality control, before presenting the event
observations. No faulty zeros or high influxes were detected during the event at any of the
eight PWS in the area of interest. Three of the PWS – PWS 1, PWS 3 and PWS 4 – were
flagged as station outliers. Nevertheless, all PWS were considered for further analysis to
compare the output of the PWS quality control with traditional evaluation metrics. The eight
PWS had between 25 and 29 stations within 10 km radius that were included in the
neighbouring checks.
The PWS time series were also checked for the full year 2022. PWS 1 and 4 were flagged as
faulty zeroes continuously during the winter months but had no Faulty Zero flags during the
summer months (see Appendix A, Figure A2). The other PWS got intermittently flagged, but
overall, there were few Faulty Zero flags during the year. No high influxes were detected at





any PWS during 2022. All stations were flagged as Station Outliers during extended periods
throughout the year, expect one (PWS 6) that only had a few Station Outlier flags in December
2022 (see Appendix A, Fig. A3).
Figure 12 shows the rainfall intensity $P$ (mm/h) observed by the eight PWS and XWR sampled
at the PWS location, with metrics calculated with XWR as reference. The correlation with XWR
was generally quite high, above 0.7 for five of the eight PWS. The CWR sampled at the PWS
locations is included for comparison. Note that the CWR time series are identical for PWS 1
and 2, PWS 3 and 4, and PWS 6, 7 and 8 respectively, meaning that the PWS are situated in
the same CWR grid cell.










***Figure 12.*** *Rainfall intensity P (mm/h) for PWS 1-8. Spearman rank coefficient $r_s$ (-) and RMSE*
*(mm/5min) calculated with XWR sampled at each PWS as reference. CWR sampled at each PWS*
*included for comparison.*
Figure 13 shows the accumulated rainfall depth $D$ (mm) for the event. Almost all PWS
significantly underestimated the total depth compared with XWR reference. However, the
estimate was closer to the reference compared with CWR, with two exceptions (PWS 5 and
PWS 7). PWS 1 is the only PWS that overestimated compared with reference, in total 7.7
mm (PBIAS -12.43%).








**Figure 13**. *Total depth D (mm), absolute difference in total depth and PBIAS calculated for PWS 1-8*
*with XWR sampled at each PWS as reference. CWR sampled at each PWS included for comparison.*

## 6. Discussion

This study investigates the capacity of second- and third-party sensors to observe short-duration extreme rainfall, compared with a conventional rainfall monitoring network. Sweden's national rainfall monitoring network, composed of automatic and manual weather stations and CWR, is used as conventional network in the study. The second-party network consists of a municipal rain gauge operated by a local water utility company and XWR. CML and PWS are studied as third-party sensors. First, a long-term analysis of the municipal gauge is performed by cross-referencing two years of data with the national monitoring network. In this way, the municipal gauge is established as a trusted reference sensor for the study. Then, a convective rainfall event that hit southwestern Sweden in the late afternoon of 18 August 2022 is selected as case study. The event analysis focuses on the urban area of Båstad, a small seaside municipality on the coast of Laholm Bay, as this location was particularly affected according to media reports.

No weather station in the national monitoring network captured the magnitude of the event as reported by media. The rainfall observed by the automatic and manual weather stations during the day of the event corresponded to a return period of less than one year, which suggests that the rainfall fell between the stations. CWR recorded a maximum total depth corresponding to a return period of 400 years, but the observation was made south of the Båstad urban area. Within the area of interest, the maximum recorded depth was only 25 mm, which does not align with the municipal observations and the media reports about flooded streets and buildings.

CWR peaked at 92 mm/h at 17:25 in the sampling point at PWS 5, which is the only CWR observation in the expected magnitude of the event based on the municipal gauge. This suggests that the CWR was affected by partial beam blockage, which has been identified to be caused by vegetation within 1 km north of the radar location (SMHI, 2020). The underestimation by CWR may also be attributed to lack of dual-polarization variables and a dedicated method for attenuation correction. Furthermore, the use of the traditional *Z-R* (reflectivity – rain rate) relationship based on Marshall-Palmer coefficients (Marshall & Palmer, 1948) in the CWR data processing may not be well suited for convective storms. SMHI is currently developing new *Z-R* relationships for different weather conditions to improve the accuracy of CWR-based precipitation estimates in the future.

When turning to the second-party data, the magnitude of the event is starting to emerge. The municipal gauge showed good agreement with the national monitoring network in the long-



term analysis and observed rainfall with a return period of around 700 years in the Båstad urban area. The XWR sampled at the location of the municipal gauge recorded a total depth of 78.4 mm, corresponding to a return period of around 800 years. It must be emphasized that estimated long return periods are highly dependent on the estimated rainfall duration, which may vary significantly in space and are difficult to firmly determine (section 5.2.1). Furthermore, return period estimates are highly uncertain and should therefore not be quantified with high precision. In this context, a difference of ~100 years must be considered relatively small and rather indicate a good agreement between the estimates.

XWR could accurately estimate the total rainfall depth compared with the municipal gauge (PBIAS 9.4%) but showed a low correlation (0.18). The low correlation may be due to differences in the observation height between radar and gauge measurements. During calmer periods of the event, rainfall accumulates gradually in the tipping-bucket gauge until a tip is registered, which can also contribute to the low correlation. XWR observations, particularly at long ranges, are known to be affected by signal attenuation due to interactions with hydrometeors. However, XWR performed well during this event at a 40-km range, likely because the event occurred locally under a mostly clear sky. Remarkably, there was no intervening precipitation between the radar and the target area. Furthermore, it is perceived that overshooting was unlikely due to higher altitude of summer precipitation compared to the XWR sampling volume at the lowest elevation angle. While near-ground radar observations are prone to partial beam blockage, this was likely less of an issue for XWR due to its larger sampling volume and steeper elevation angle compared to CWR, which observed the event from a much shorter distance (6 km).

One CML with a length of 4.8 km is located in the area of interest. The CML correlated well (Spearman coefficient 0.9) with XWR and observed similar duration of the event, which is promising for binary rain/no rain detection. However, the total depth was underestimated compared with the reference as the CML rainfall rate reached a 'plateau' and stayed constant at this level for about 30 minutes. This effect is sometimes referred to as 'blackout' (Polz et al., 2023) and appears when the radio signal is completely attenuated by heavy rainfall (ITU-R, 2005). Telecom network providers design the CML hardware so that transmission outages are allowed to occur 0.01% of the time on an annual basis. Indeed, Polz et al. (2023) found that blackout gaps were present in less than 1% of attenuation data from 4000 CMLs over 3 years in Germany, and that the effect on long-term timescales was generally low. However, the probability of a blackout at rainfall intensities above 100 mm/h was above 40%, which implies that the CML technology currently has limitations in quantifying extreme events.



The analysis of XWR data along the CML link revealed some notable result. Firstly, the XWR
data at some time steps exhibited a large bin-to-bin variability, sometimes shifting from one
intensity level to another (Fig 10b). This can be attributed to the turbulent nature of convective
storms, and local attenuations of XWR signals during heavy rain bursts. Despite overall
agreement between $\bar{P}_{XWR}$ and $\bar{P}_{CML}$ along the link, a substantial scatter was found where, in
particular, low intensities were consistently higher in the XWR data than CML. Generally, there
was a clear indication that the CML underestimation increased with increasing rainfall intensity
as well as variability along the link. The underestimation observed by both CWR and CML
compared with XWR may partially be explained by XWR's shorter wavelength which interacts
better with small hydrometeors (Lengfeld et al., 2016). Notably, the estimations of the event
duration based on radars and CML was significantly different from the in-situ gauge
observations. For example, the municipal gauge started to observe the event 30 minutes after
CWR. These discrepancies could be attributed to the influence of wind on aloft observations,
larger sensitivity of CML and radars to light rainfall and slow accumulations in the tipping-
bucket gauge during light drizzles preceding the heavy bursts.
Regarding the eight PWS in the area of interest, the tipping bucket mechanism seem to have
reached a maximum frequency during heavy moments of the event, as no observation
exceeded 100 mm/h. A similar tendency has been observed by others (Lussana et al., 2023;
Wolf & Larsson, 2024). Among the PWS with lowest RMSE, this led to a PBIAS of 30-40%
compared with XWR reference. PWS 1 performed reasonably well on all evaluation metrics
with a Spearman correlation of 0.9, RMSE 2.2 mm/5min and PBIAS −12.43%. In most cases,
the correlation with reference was medium to high, with only 2 PWS with correlation below 0.6.
We applied a quality control specifically designed for PWS rainfall data, *pypwsqc* (Chwala et
al., n.d.) on the event and full year 2022. The algorithm applies three filters – Faulty Zeroes
filter, High Influx filter, and Station Outlier filter – to assess the quality of each time step by
utilizing neighbour checks with nearby stations. No faulty zeroes were detected during the
event, which is reasonable as all PWS in the area of interest measured rainfall at all timesteps.
No high influxes were found, suggesting that all PWS in the area measured enough rainfall not
to trigger high influx flags at the neighbouring stations. On the other hand, no high influx was
detected at any PWS during the entire year 2022. There might indeed not have been any high
influx recorded by any of the 58 PWS on the Bjäre Peninsula in 2022 but the results also raise
the question if the filter parameters should be tuned differently to better capture unrealistically
high inflows.
Regarding the Station Outlier filter, three stations were flagged as station outliers during the
event – PWS 1, PWS 3 and PWS 4. However, when inspecting the time series and evaluation
metrics for these stations, it appeared that PWS 1 and PWS 3 had among the highest



correlation and lowest RMSE of all PWS and generally showed a reasonable rainfall pattern
compared with the other PWS. These results point to a limitation of neighbouring checks in the
context of convective storms. PWS 1, PWS 3 and PWS 4 are all located in the western part of
Båstad. As such, the Station Outlier filter considered the observations of PWS located further
to the west on the Bjäre Peninsula, which experienced a total depth of only 10 mm according
to the XWR observations. The high spatial variability of the event therefore triggered station
outlier flags at the three PWS located closest to the drier area, even if two of them performed
well when comparing with the XWR reference. If the flagged PWS would have been removed
from further analysis based on the results from the Station Outlier filter, sound observations
would have been lost. Conversely, the performance of PWS 5 and PWS 7 was very poor
compared with the XWR reference, but these stations were not flagged in the automatic quality
control. Future research could explore how the spatial density of PWS and considered
evaluation range influence the capability of neighbour checks to be applicable as quality control
protocols for localized rainfall.
The findings of this study align with the well-established fact that national monitoring networks
have limitations in terms of observing convective rainfall. To strengthen capacity in this field,
national meteorological services can include second-party data in operational tools and
workflows. However, differences in acquisition protocols, data formats etc. adopted by different
actors may cause an additional burden and hinder the integration of second-party sensors.
Importantly, southwestern Sweden has an excellent coverage of second-party sensors thanks
to the combination of XWR and rain gauges operated by local authorities, which is certainly
not the case for all points of interest, particularly in countries with limited resources (Winsemius
et al., 2018). In those cases, national meteorological services can turn to third-party sensors,
particularly CML that are typically available in populated settlements across the globe (Chwala
& Kunstmann, 2019; Blettner et al., 2023). However, the results of this study suggest that these
sensors currently have limitations in quantifying the correct magnitude of convective storms.
Still, the results show that third-party data may assist in detecting storm durations and binary
rain/no rain detection.
Regarding limitations of the study, a few remarks can be made. First, there are uncertainties
associated with all observations in the study, especially the indirect rainfall measurements
(radars and CML) and the PWS. The long-term assessment of the municipal gauge, combined
with the good agreement between the municipal gauge and XWR, still provide solid evidence
for the actual magnitude of the event. Secondly, some findings are expected to be specific for
this study, such as the low performance by CWR caused by beam blockage in the area of
interest. On the other hand, the underestimation of rainfall observed by the third-party network
aligns with previous studies. It is also expected that quality control protocols that utilize




neighbouring checks will be problematic for other convective storms, depending on the station
network density and considered range of the analysis. Although no general conclusions can
be drawn from a case study, the depth of this analysis contributes to the understanding of
advantages and limitations when observing convective rainfall with second- and third-party
sensors.

## 7. Conclusion

This study investigated the capacity of second- and third-party sensors to observe short-
duration extreme rainfall compared with a conventional rainfall monitoring network in a case
study. The results show that the conventional network was unable to fully capture the event
and that second-party sensors can provide accurate and detailed representations of
convective storms. However, second-party sensors are not always available, particularly in
resource-strained settings. Furthermore, the results suggest that third-party sensors can
assist in detecting storm durations and spatial variability of rainfall but have limitations in
quantifying the correct magnitude of convective storms.  Third-party data may also be difficult
to obtain for national meteorological services and has known problems with data quality.
Future research is suggested to continue the efforts on quality control of third-party data,
especially related to extreme events. In addition, more research is needed on the integration
of second- and third-party data in the workflows of national meteorological services.

## Appendix A

### A.1 CML processing

The MEMO (Microwave-based Environmental Monitoring) method was developed and tested
on an open data set ('*OpenMRG*') that consists of 364 CML and 11 rainfall gauges in
Gothenburg, Sweden, for the period June-August 2015 (Andersson et al., 2022).   The
processing steps of the MEMO methodology are outlined below.

### A.1.1 Data pre-processing

The 10-second attenuation was calculated by taking the difference between TSL and RSL.
Then, the median value over a 1-minute period $A_{ml}$ was taken for all minutes that had more
than four 10-second values in one minute and if less data were available, that minute was
flagged as missing data.

### A.1.2 Wet-dry classification

Sub-links in the *OpenMRG* dataset were scrutinized to find a wet-dry classification method that
does not rely on secondary observations. The links considered were located within 500 m from
a municipal rain gauge in Gothenburg that records at 1-minute temporal resolution, resulting
in 72 links. First, dry time steps recorded by the station between 2015-05-14 to 2015-08-31
were considered. A time buffer of 30 minutes was added before and after each rain event



recorded by the rain gauge, to consider that rainfall arrives at different timesteps to the links.
The 99th percentile of $A_{ml}$ at dry timesteps identified by the rain gauge was considered to
address that the links may record rainfall that was missed by the rain gauge. Then, the
empirical distribution of $A_{ml}$ at the dry timesteps was plotted and inspected for the 72 links. An
example is shown in Fig. A1. In this example, the difference between the median and 99th
percentile of the attenuation is 0.35 dB.

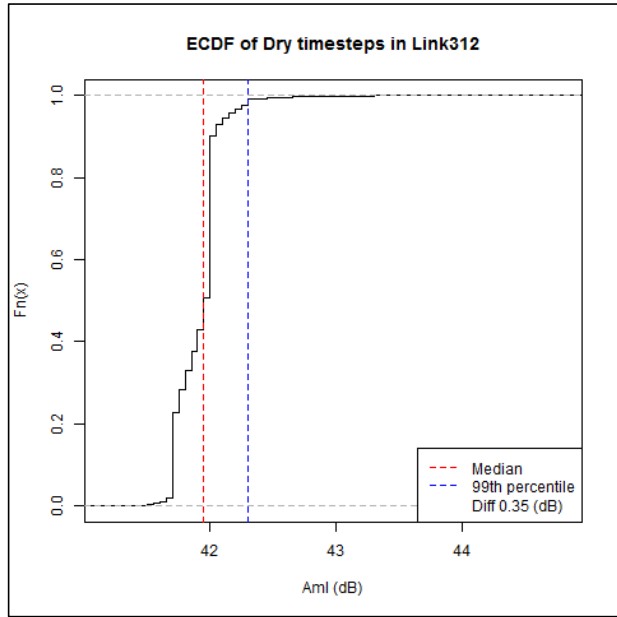


**Figure A1.** *Example of empirical distribution of attenuation level ($A_{ml}$) at dry timesteps for Link 312.*
The plots showed that the difference in $A_{ml}$ between the median attenuation and the 99th
percentile was typically between 0.35-0.6 dB at dry timesteps. However, the difference for one
link with considerable fluctuations in signal attenuation was 1.7 dB. Based on these results, it
was decided to set the threshold for the wet-dry classification to the median attenuation over
the past 2 weeks plus an additional 1.7 dB (here called the '*median buffer method*'). In this
study, where only two days of data was available, the median was taken over all available
preceding time steps.
The median buffer method was compared with classifying all timesteps with attenuation above
the median of the last two weeks as wet ('*median method*') and the method presented by
Schleiss and Berne (2010) ('*Schleiss method*'), see section 4.4.3. The median method resulted
in overestimation of the number of wet timesteps compared with the rain gauge. The Schleiss
method performed similarly to the median buffer method in correctly identifying the number of



wet timesteps but resulted in some outliers and produced more false wet time steps. Based on
these results, the median buffer method was used for further analysis.

### A.1.3. Baseline definition

The baseline $A_{bl}$ is the expected difference between TSL and RSL during dry weather. This
means that during dry periods, based on the wet-dry classification in the previous step, the
baseline is equal to the attenuation $A_{ml}$. During wet periods, the baseline is taken as the
median of the last $N$ timesteps from the first wet timestep. A suitable reference period for $N$
was found to be 240 minutes.

### A.1.4. Conversion of net attenuation to rain rate

By subtracting the baseline from the attenuation, the net attenuation $A_{nl}$ was found as

$$A_{nl} = A_{ml} - A_{bl} \tag{A1}$$

Following common practice in CML literature (Leijnse et al., 2007b; Messer et al., 2006),
attenuation was converted to rain rate $P_{raw}$ using the power-law relationship:

$$A_{nl} = k P_{raw}{}^{\alpha} \tag{A2}$$

The same relationship is used for radar processing, see Eq. 7. In radar literature, $A_{nl}$ is often
expressed as the one-way specific attenuation, $k$ (dB/km).
The parameters $k$ and $\alpha$ depend on link frequency, the polarization state, and the elevation
angle of the signal path and was found by applying the equations derived by ITU-R (2005). In
contrast to radar scatter, the sensitivity to DSD (Eq.1) is very limited around 30 GHz because
$\alpha$ is approximately 1 in this range, suggesting a nearly linear relation between net attenuation
and rain rate (Chwala & Kunstmann, 2019). At frequencies further from 30 GHz, DSD will play
a larger role and biases can occur. Most links in Sweden operate near 30 GHz (Andersson et
al., 2022).

### A.1.5 Bias correction based on link length

The derived rain rate was analyzed for the 72 links situated within 500 m range from the 11
rain gauges in the *OpenMRG* dataset for July 2015. When plotting the residuals of the rain rate
at the closest gauge against 15-min accumulated net attenuation of the link, a linear
relationship was found, indicating potential for bias correction. The slope of the residuals was
derived by linear regression for each link and plotted against the link frequency, link length and
the parameters $k$ and $\alpha$ in Eq. A2. The most distinct relationship was found for link length,
suggesting the shorter the length, the higher the slope of the residuals. One probable reason
for the relationship is the wet-antenna effect, which is stronger over shorter distances (Chwala
& Kunstmann, 2019).



It was found that the slope of the regression line of the residuals could be estimated from link
length by applying a simple inverse equation:

$$Slope = f \times \frac{1}{L^g} + h \tag{A3}$$

where $L$ is the link length. The parameters $f$, $g$ and $h$ were optimized by minimizing the Mean
Absolute Error for the 72 links, arriving at 2.85214, 1.672 and 0.1615, respectively. The bias
corrected rain rate for the CML in Båstad was then found by calculating the correction factor:

$$CF_A = 2.85214 * (1/L^{1.672}) + 0.1615 \tag{A4}$$


where $L$ is 4.8 km in this case. Then, applying the factor to the derived rain rate:

$$P_{CML} = P_{raw} - (A_{nl} * CF_A) \tag{A5}$$


## A.2 PWS quality control 2022
Figure A2 shows Faulty Zero (FZ) flags for the eight PWS in the area of interest for the full
year 2022.



**Figure A2.** *Faulty Zero (FZ) flags 2022. 1 = FZ flag, 0 = no FZ-flag, -1 = FZ-filter could not be applied.*





Figure A3 shows Station Outlier (SO) flags for the eight PWS in the area of interest for the
full year 2022.






**Figure A3.** *SO-flags 2022. 1 = SO flag, 0 = no SO-flag, -1 = SO-filter could not be applied.*



## Data availability

National weather station data and C-band radar composite are available from SMHI's open
data archive (www.smhi.se/data). The processed national data are available on request.
Data from the municipal gauge and X-band radar is property of Nordvästra Skånes Vatten
och Avlopp, NSVA. CML data is property of the telecom companies Ericsson AB and Tre.
PWS data is property of Netatmo. However, Netatmo data can be openly accessed through
their API: https://weathermap.netatmo.com

## Author contribution

LPW processed gauge data. LPW and RvdB processed CWR data. HaH, LPW and RvdB
processed XWR data. RvdB and LPW processed CML data. JA developed the CML processing
methodology. LPW implemented the PWS quality control. LPW, JO and HaH performed data
analysis. All authors were involved in the writing and editing of this manuscript.

## Competing Interests

The authors declare that they have no conflict of interest.

## Acknowledgements

The authors would like to thank NSVA for sharing tipping bucket data and Ericsson AB and
Tre for sharing CML data. Lennart Simonsson processed PWS data to NetCDF-files. Riejanne
Mook improved the CML processing code. The development of *pypwsqc* was carried out jointly
with Christian Chwala within the working group 2 of the COST Action OpenSense CA20136.

## Financial support

The paper was supported by the projects SPARC (*Stakeholder participation for climate
adaptation – data crowdsourcing for improved urban flood risk management*, grant 2021-
02380 Formas) and HyPrecip (*Hyper-resolution multi-dimensional precipitation information
for analysis and modelling: urban and rural applications*, grant 2021-01629 Formas). The
development of *pypwsqc* received funding from the European Union's Framework
Programme for Research & Innovation as part of the COST Action *OpenSense* [CA20136],
as supported by the COST Association (European Cooperation in Science and Technology).



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

professionell-anvandning-test-av-netatmo-regnsensor-i-trelleborg-och-svedala/
