# Peer review of "Have you ever seen the rain? Observing a record"

_EGUsphere, 2025_

## Author Response (AR1)

2025-11-21
Louise Petersson Wårdh
louise.petersson.wardh@smhi.se

**Response file**

This document contains comments from reviewer 1 and 2, our responses and changes in the manuscript. The changes are referring to line numbers in the **marked-up manuscript with track-changes.**

**Reviewer 1**

**Comment 1**

Line 161: I was unsure what area this portion of the text was referring to in Figure 1. Is this just a composite of SMHI stations? Clarity could be improved here so readers understand the monitoring locations.

Response: Thank you for pointing this out. The text is referring to the C-band radar located 6 km South of the study area, marked with a white dot in Map 1 in Figure 1. The radar location is unfortunately missing from Map 2 in Figure 1 and will be added.

Changes: The location of the C-band radar has been added to map 2 in Figure 1.

**Comment 2**

Line 194: Can you define overshooting in the text? What does it mean in this context?

Response:

Overshooting in this context means that the radar beam shoots above the precipitation cloud and hence does not record it. It is a common source of error in radar data. We will add some standard literature as reference for further reading on the topic. The purpose of this section is to see if there is risk of overshooting by the X-band radar in the area of interest. The same analysis is done in lines 168 – 171 for the C-band radar. As the radar beam that we are using travels on 200-300 m height at the area of interest for CWR and 300-1200 m for XWR, and convective precipitation in the summer months in Sweden typically originate from much higher altitudes, the risk of overshooting is very small.

Changes: Definition and literature added to lines 198-200.

**Comment 3**

Line 227-229: This sentence would benefit from more clarity and further explanation.

Reply: the intention of this sentence is to add clarity, but if it only confuses things, it is maybe better removed. For sub-hourly rainfall data, the correlation between time series recorded by nearby sensors can be very low even if they record similar total rainfall depths (for example Spearman correlation 0.18 between XWR and the municipal gauge, see Results). As convective rainfall is highly variable in space and time the observations per time step can be very different at nearby locations. If you use correlation as metric this will indicate poor performance, when, in fact, the sensors simply may experience different rainfall intensities even if they are closely located. By shifting the time series in time, you can account for the fact that a certain rainfall intensity may be observed by the radar before it reaches a rain gauge on the ground, for example.

Changes: Lines 295-299 have been removed. This content is now described in more detail in section 4.5.

**Comment 4**

Line 272 and 274: I appreciate the clear explanation of all equations and understand that several of the parameters were obtained from other sources. Could you briefly elaborate what the G and offset variables represent for Equation 6 as well as a and b for Equation 7?

Response: Reflectivity data from radars are stored and distributed as integers between 0 and 255 to enable smaller storage size, following European standards (Michelson et al., 2014). To convert these integers back to reflectivity (dBZ) you apply the coefficients G (gain) and offset.

a and b in equation 7 are well-established empirical constants that Marshall and Palmer (1948) found when establishing the relationship between size distribution of raindrops in a radar pulse volume to the rainfall rate.

Changes: Clarification added to lines 178-187. The content has been moved to section 3.1.

**Comment 5**

Line 392: Is it typical to see a trend with higher accumulated depth at inland regions like Baramossa? A reference here may be appropriate to establish that your results are typical and expected.

Reply: This is not a very general feature for Sweden, but the statement referred to the specific study area. The pattern can be seen in maps of 30-year mean annual precipitation here: https://www.smhi.se/klimat/klimatet-da-och-nu/normalkartor/normal/arsnederbord-normal. We will clarify this part of the text in the revised manuscript.

Changes: Clarification and reference added to lines 569-572.

**Comment 6**

Figure 4: It's difficult to tell the difference between SMHI Hov and SMHI Laholm D. The line types are very similar. This figure would be more clear with a different line choice for one of those data sets.

Reply: Thank you for pointing this out. The line types will be better differentiated in the next version.

Changes: New line styles for SMHI Hov and Laholm D in Figure 4 have been added.

**Comment 7**

Line 494: Could you elaborate in the text on why the plateau period was considering unsuitable for comparison?

Reply: Given the sudden constant records of rainfall rate observed in the CML time series, which is unexpected and clearly not representative for the actual rainfall rate, it was excluded from the analysis.

Changes: Clarified in lines 692-693.

**Comment 8**

Figure 10: Is each panel a-e correspond to one minute? For example: panel a is 17:22-17:23 and panel b is 17:23-17:24. If so, this is not clear based on the information provided in the figure caption.

Reply: Thank you for pointing this out. It is correct that, for example, panel a is 17:22-17:23 and panel b is 17:23-17:24. This will be clarified in the next version.

Changes: Time stamp added to the panels in figure 10.

**Comment 9**

Figure 11a: Are you expecting a linear trend with this comparison? If so, reporting an $R^2$ value would be useful.

Reply: Thank you for pointing this out. A linear trend is expected because, ideally, the two sensors (CML and XWR at the CML location) should record the same or very similar rainfall intensities. $R^2$ will be reported in the next version.

Changes: $R^2$ added to figure 11a.

**Comment 10**

Figure 12: The figure caption, please specify that the $r_s$ is relating the data to XWR data.

Reply: Thank you for this valid and helpful comment, it will be added in the next version.

Changes: Specified in the caption of figure 6, 8, 9, 12.

**Comment 11**

The discussion overall is thorough and touches on many good points. However, is there a reason the results and discussion sections are separate? At times this made assessing the claims made in the discussion challenging due to needing to locate the appropriate data in the Results section. If you choose to leave the Discussion as a separate section, please refer the reader to relevant tables and figures when emphasizing trends observed in the data. I have noted a few specific lines below, but I encourage you to go back through this section for any crucial information that would benefit from being redirected back to a table or figure.

Lines 589-591, Lines 598-599, Line 604, Line 612-616, Lines 628-630, Lines 644-645, Lines 660-662.

Reply: We were indeed discussing whether to keep the results and discussion sections together or apart before arriving at this solution. It is understood that the discussion, if kept separate, will need more cross-references to the text. We will await the second review to decide which approach to choose.

Changes: More references to figures and sections have been added throughout the Discussion section.

**Reviewer 2**

2025-11-21
Louise Petersson Wårdh
louise.petersson.wardh@smhi.se

**Comment 1**

Line 87: CMLs measure path-integrated attenuation, not rainfall. I guess that is what you're trying to say with this sentence: that the assumption is that this is directly related to the path-averaged rainfall intensity.

Response: Thank you for noticing this – "rainfall" will be replaced with "attenuation".

Changes: The sentence has been removed, to address comment #2.

**Comment 2**

Lines 88-89: this assumption has been investigated by Berne et al. (2007), Leijnse et al. (2008, 2010), and De Vos et al. (2018) in high-resolution simulation frameworks. It would be interesting to see how your results for this case compare to what is presented in this literature.

Response: We will refer to these papers and reflect on how the results compare with this literature in the discussion.

Changes: Lines 89-95 in Introduction and 853-855 in Discussion.

**Comment 3**

line 94: consider using "people" instead of "occupants".

Response: We will replace "occupants" with "people".

Changes: "Occupants" has been replaced with "people" in line 96.

**Comment 4**

line 107-108: you state that "cross-referencing radar observations with path-integrated rainfall estimated by CML" is a gap in the field of high-resolution rainfall monitoring. A large portion of the work cross-references radar and CML rainfall estimates, so I don't fully understand this statement. Could you make the statement more specific about which gap exactly is bein addressed here?

Response: We are specifically referring to line 89-93 and the results in section 5.2.4, especially the visualization in Fig. 10 and the analysis in Fig. 11. However, we acknowledge that this has been addressed in Leijnse et al. (2008, 2010) and other studies and we will therefore remove it from the list of research gaps and rephrase line 89-93.

Changes: Lines 89-95, lines 108-112.

**Comment 5**

lines 109-114: I think that the research questions as they are formulated here cannot be answered by the analysis of a single convective event, because it probably depends greatly on the path of the storm and the local topology of the sensors in that path. Could the research questions be reformulated to take this into account?

Response: These are guiding research questions and we don't claim to answer them completely once and for all with this study. We will relax the framing of the questions even further to emphasize this.

Changes: The framing of the research questions have been changed in line 113. However, we think the questions themselves are fine as guiding research questions.

**Comment 6**

Figure 1: it would be instructive to have different symbols on the map for the SMHI automatic gauge and the SMHI manual gauges.

Response: This will be changed in the figure.

Changes: The automatic and manual stations have been differentiated in map 2 in Figure 1.

**Comment 7**

Figure 1: could the location of the C-band radar be put on map 2 as well, so that it is clear where the radar is relative to the study area?

Response: The location of the C-band radar will be added to map 2.

Changes: The location of the C-band radar has been added to map 2 in Figure 1.

**Comment 8**

Figure 1: would it be possible to add the predominant wind direction to the figure? This could help in interpreting differences between the municipal gauge and the SMHI gauges.

Response: We are not sure if the reviewer asks for the predominant wind direction over all in this region or for the studied event specifically. In any case, the answer in both cases is southwest. This will be added to the text.

Changes: Added in text to line 138-139.

**Comment 9**

Line 155: it is mentioned here that "gauge-adjusted Plan Position Indicator" is used. However, in the subsequent paragraphs there is no information about how the gauge adjustment was carried out. Because this is very relevant for interpreting the results, please add information about this in the paper. Furthermore, I get the impression here that a precipitation product is used in this paper, but from Section 4.4.1 it seems that it is actually a much more raw product. Please clarify this in the paper. And if this is indeed a more raw reflectivity product, then I would suggest to use the radar data on its native (spherical) grid, i.e., not the 2x2 km composite because this would give much more spatial detail.

Response: The gauge adjustment technique is described in Michelson & Koistinen (2000) and we will outline the method briefly here. It is mentioned in lines 164-165 that the composite is distributed as reflectivity data, we will move this a bit earlier in the text for clarity. The composite must indeed be processed as described in section 4.4.1. Please refer to response #13 below for more details on the selection of this composite. There is no precipitation product available for CWR, as is the case with XWR in this study.

Changes: Lines 167-170, section 3.1

**Comment 10**

Line 155: consider adding "reflectivity" (or even "horizontal reflectivity") between "radar" and "composite" to make clear what kind of PPI is used.

Response: The text will be changed to "radar horizontal reflectivity composite"

Changes: Text updated in lines 163-164.

**Comment 11**

Lines 157-160: here you explain Z in terms of the DSD (you could also have chosen to present it in the form of the radar equation). To me it seems more relevant to present the resulting Z-R relation instead, preferably the one that was used to retrieve rainfall intensity for this study.

Response: We introduce the fundamental concept of DSD here and the applied Z-R relation in section 4.4.1 (methods).

Changes: No changes made to address this comment specifically, but the content of section 4.4.1 has been moved to section 3.1 instead, which we think creates better readability of the radar equations.

**Comment 12**

lines 161-162: the degree to which a radar contributes to a given point in a composite depends on the employed compositing method. So consider adding "and the compositing method is based on the closest radar" (or something similar, depending on the compositing method that is used) after "during the selected event".

Response: The text will be updated as suggested in the comment.

Changes: "The compositing method is based on the closest radar" has been added to lines 173-174.

**Comment 13**

Line 163: here it is mentioned that the resolution of the composite is 2x2 km. Given the very close range and the expected high spatial variability of the precipitation, using the radar data on its native (spherical) grid would give much more spatial detail of the storm. Can you indicate the reason for using the composite in the paper?

Response: The main reason for using this composite in the paper is reproducibility; it is the only SMHI radar product that is distributed openly. Secondly, the purpose of the paper is to benchmark second- and third-party sensors with a "conventional monitoring network", not with the "best available radar product". The applied composite is used operationally by the forecasting service at SMHI and distributed to external partners that use the radar sequences in decision making. We therefore consider this product to be the best representation of a "conventional monitoring network" in Sweden.

Changes: Addressed in line 165.

**Comment 14**

Lines 171-173: here you discuss partial beam blockage. Could you indicate the severity of the beam blockage? And could the location of the vegetation causing this blockage be indicated on map 2 of Fig. 1? This is relevant for the interpretation of the results.

Response: When studying accumulated total depth recorded by this radar for the years 2022-2023, an internal evaluation at SMHI showed that the radar underestimated the accumulated depth compared with the SMHI rain gauges located to the north of Båstad with about 80%. The beam blockage affects a circular sector of 60 degrees to the North of the radar location. We don't find it appropriate to add the location of the vegetation to Figure 1 as the reader is not introduced to the problem of beam blockage by the radar at that point in the text. The affected area is shown in the reference SMHI (2020). This webpage also includes a north-south cross section (note that the area of interest in the study is located straight to the north of the radar location) which shows the elevation and vegetation in the area in relation to the radar beam, clearly indicating that the main lobe is blocked by vegetation. These figures will be added to the appendix.

Changes: Figures added to appendix (section A1) and the implications of this have been given more attention in the discussion section (lines 787-791).

**Comment 15**

Line 190: dual-pol attenuation correction is mentioned here, and the reader is referred to the literature for details on the method. I think it would be relevant here to add a few words (or a sentence) about which method is used (referring to the literature for details of the method is fine).

Response: This method estimates attenuation as an approximately linear function of the specific differential phase shift, which depends on phase rather than signal intensity and is therefore less sensitive to attenuation (Kumjian 2013). We will mention this in the text and refer to Hosseini et al. (2020) for further details.

Changes: Clarifications added to lines 231-235.

**Comment 16**

Line 191 and Figure 3: "half-beam" and "half beamwidth" are mentioned here. What is meant by this? Do you mean half-power beam width? If so, please modify this. And if this is the case, then consider to use the same terminology for the C-band radar in Fig. 2.

Response: Thank you for this question. We noticed an error in Fig. 3. For consistency, we will update both Fig. 2 and Fig. 3 so that we illustrate the beamwidth for both radars. To clarify, these are the nominal beamwidths provided by the manufacturers, sometimes referred to as half-power or –3 dB beamwidths.

Changes: Figure 2 and 3 have been updated and changed to display full beamwidth for both radars.

**Comment 17**

Lines 198-204: what is the frequency of this link? Please also indicate this in the paper.

Reponse: The frequency is 23.1 GHz, this will be added to the manuscript.

Changes: Link frequency added to line 251.

2025-11-21
Louise Petersson Wårdh
louise.petersson.wardh@smhi.se

**Comment 18**

Section 4.1: the evaluation metrics are discussed here. You choose to use the Spearman rank correlation over the more traditional Pearson correlation. I'm assuming that the reason for this is the insensitivity to outliers. Please indicate the main reason for using this correlation parameter. And you're also using the RMSE to complement the analysis, which I fully support. Consider using a normalized version of the RMSE (normalizing by the mean reference precipitation, like you do for PBIAS in Eq. (4)).

Response: We indeed think Spearman rank is a better choice in the context of convective storms due to the large variability and range of observations during such events. It also captures non-linear relationships. We considered to use normalized RMSE but will instead change the unit to mm/h in all plots (see comment 35 – thanks for pointing this out). This will allow intercomparisons both between different plots and with the y-axis in each plot.

Changes: RMSE expressed as mm/h in Fig 6-12.

**Comment 19**

Section 4.1: on what time resolution are the metrics computed? Is it the native resolution of the sensor with the coarsest resolution? Or do you use a common time resolution for all sensors? The resolution can have a large impact on the values of the metrics, so it is important to know, and to take into account in interpreting the results.

Response: The metrics are indeed calculated on the resolution of the sensors with the coarsest resolution – PWS and CWR – which is 5 minutes. So, the metrics are calculated on 5 minutes resolution (except the long-term analysis, which is daily). This will be added to the text.

Changes: The content has been moved from section 4.1 and incorporated in section 4.2 and 4.3 instead as there are different time resolutions of the metrics. We think it is clearer to introduce it where each comparison of data is described than in section 4.1.

**Comment 20**

Lines 225-227: a time shift is discussed if correlations are low. I think this is a good idea. However, it is not clear from the paper when exactly this time shift is applied ("If very low correlations (close to 0) were found" is too vague). And it is also not clear how the time shift is determined. This should be included in the paper. Is the same time shift also applied when computing the RMSE? Please include that in the paper, too.

Response: Thanks for pointing this out. We will add a section in Methods called "Time lags" or the like where the procedure will be described. In addition, the lags have been recalculated and new (more reasonable) numbers will be reported- this affects comment #34 and #38 as well. See response #34 for more details on the new results.

The time lags were calculated on 5 minutes resolution, as this is the resolution of the CWR data. Time lags from -10 (shifting radar 10 timesteps back in time compared with the municipal gauge) to 10 timesteps (same procedure but ahead in time) were considered and for each lag the Spearman rank correlation was calculated over the event duration as defined by the municipal gauge. The best correlation value was then selected. It should be noted that the selection of event duration has a

great impact on these calculations. Including just one extra time step can give very different correlation values, typically because it introduces more zero-rainfall-measurements and hence spuriously increases the correlation – we therefore find it important to keep section 5.2.1 in the paper to be clear on the event duration for each sensor (see comment 33).

In fact, time lags were only applied to the radar data. The idea of applying time lags appeared when "low" correlations between the radars and municipal gauge were found. For the opportunistic sensors, the lowest correlation found was 0.44 compared with reference (PWS 4). For the radars, which we would have expected to perform better, the original calculations showed a correlation of 0.07 (CWR) and 0.18 (XWR) compared with reference. However, these results were based on incorrect duration of the storm (see response # 34). Rather than setting an arbitrary correlation threshold for the application of time lags, we will just state that time lags were applied to the radar data in the new section.

The RMSE (and all other metrics) is calculated on the original time series, not the shifted ones. This shows how the sensors perform before applying lags which we think is more in line with the scope of the paper. This will be clarified in the results section.

Changes: Section 4.5 (Time lags) have been added.

**Comment 21**

Section 4.4: I found myself scrolling back to Section 3 very often when reading Section 4.4. I therefore think it would greatly improve the readability of the paper if the contents of this section are moved to Section 3, where the different datasets are introduced.

Response: We agree and will incorporate the content of section 4.4 to section 3 instead.

Changes: Most content of section 4.4. (Methods – Data processing) has been moved and incorporated in section 3 (Data) instead. However, some content on data processing that did not fit under section 3 is still in section 4.3 – 4.5.

**Comment 22**

Lines 269-272: I think this text is not relevant for the paper. These are technical details of how to extract Z from the radar files and can be omitted from the paper in my view.

Response: We think it is relevant to report which coefficients that were used to convert reflectivity to rain rate for reproducibility of the results and will therefore keep these lines in the text.

Changes: None. Note that this content has been moved to section 3 (Data) instead.

**Comment 23**

Lines 277-281: it is not relevant that the resulting data are stored as geoTIFF files. You could also just summarize this paragraph by saying that CWR time series at a 5-minute resolution were created at the locations of the municipal rain gauge an the eight PWS.

Response: The text will be updated accordingly.

Changes: Changes have been added to lines 184-187. Note that this part of the text has been moved from section 4.4 to 3.1.

**Comment 24**

Lines 294-295: it is indicated here that there were missing values in the X-band radar data during the most intense part of the storm. This in itself is relevant information about the usefulness of such data: if data are missing when they are most crucial then the radar becomes much less useful. Therefore it would be very useful to know why these data are missing. And are entire 1-minute data missing, or are there gaps in the data around the most intense part of the storm? Please elaborate on this in the paper.

Response: We agree that this is important to highlight this to understand the applicability of XWR for monitoring of convective storms. There are individual bins missing data around the intense part of the storm, not the full domain. Investigations suggest that this not due to a limitation of the sensor itself but imposed by post-processing in the pre-calculated rain rate that we used for the analysis. The data was compressed to 8-bit format and therefore any value above 255 (mm/h in this case) was lost. VeVa (see line 186) has since then changed the compression to 16-bit which allows for observations up to 650 mm/h without losing data, but the error is unfortunately present in 2022 data. We will elaborate on this in the paper.

Changes: Lines 345-349.

**Comment 25**

Line 295: linear interpolation is mentioned here. Is it spatial or temporal linear interpolation. Please mention this explicitly.

Response: It is temporal linear interpolation, which will be clarified in the text.

Changes: Line 349.

**Comment 26**

Lines 295-298: what is the reason for regridding onto a 500-m grid here? I think you can lose a lot of detail by doing this (given the native resolution of 250 m in range and approximately 350 m in azimuth at 40 km range).

Response: All XWR time series in the paper are sampled directly from the polar bins as described in lines 288 to 292. The data was resampled to a cartesian grid for the following reasons: 1) create the map in Figure 7, 2) to get a spatial overview of the total depth of the event and (importantly) 3) to allow for comparison with the CWR composite in Figure 5. We believe a resolution of 500 m serves these purposes. Although a higher spatial resolution of (for example) 250 m would have given even more detail it would also increase the difference even more to the CWR resolution. We will update the text to clarify this.

Changes: Lines 341-344 have been updated for clarity. Note that the native resolution is 75 m, not 250.

2025-11-21
Louise Petersson Wårdh
louise.petersson.wardh@smhi.se

**Comment 27**

Section 4.4.3: CML processing is first discussed in terms of a short literature overview of methods, after which the reader is referred to the appendix for information about which methods have actually been used in this paper. I think the literature overview of the methods (lines 301-319) should be moved to the appendix (it is valuable information, but it distracts from the main topic of the paper). The methods that are used in this paper should then be briefly described.

Response: We will move the literature review to the appendix and more content on the CML processing method will be moved from the appendix to this section.

Changes: Lines 301-319 in original manuscript has been moved to the appendix. Lines 259-266 have been updated to reflect the comment.

**Comment 28**

Line 329: a difference in total rainfall depth of 3 mm is judged to be small here. The reader has not yet come across numbers of total accumulation as measured by the CML, so it is difficult to judge whether this is indeed the case. I suggest either adding the total accumulation for the links here, or expressing the difference as a percentage of the total accumulation.

Response: We will express the difference in percent instead. The difference is 3% (the total depth recorded by the sub-links was 57 and 60 mm, respectively).

Changes: Difference as percentage added to line 351 (the difference is, however, 5.1%).

**Comment 29**

Line 335: it is stated that "XWR bins were sampled each 250 m". However, in Section 4.4.2 (lines 295-297) you explain that the "volumetric data was gridded into a cartesian grid [with a resolution of] 500 meters". So sampling it at 250 m doesn't make sense to me. The effective spacing of the data is then between 500 m and 700 m, depending on the orientation of the grid with respect to the CML. I think this should be made clear in the paper. And an explanation should be added about the reasons for this way of sampling the XWR data.

Response: as outlined in response 26, all XWR time series in the paper are sampled from the polar bins as described in lines 288 to 292, not from the 500 m grid. That grid was created for the reasons outlined in response 26, mainly for visualization purposes in Fig 7.

Changes: Lines 341-344. Note that the native resolution is 75 m, so in the area of interest the XWR bins can be approximated as 75x350 rectangles.

**Comment 30**

Line 344: how relevant for this paper is the fact that the data were available as csv files and then converted to netCDF? I think this could be removed.

Response: The text will be updated to reflect the comment.

Changes: The text has been deleted.

2025-11-21
Louise Petersson Wårdh
louise.petersson.wardh@smhi.se

**Comment 31**

Table 1: if these numbers are presented here, then the meaning of all of the variables should be made clear in the text. Alternatively, you could remove the table, and only provide the values of $m_{match}$ and $m_{int}$ (the fact that you're using the values from de Vos et al. (2019) is already in the text on lines 375-376).

Response: The table will be removed and the $m_{match}$ and $m_{int}$ values will be described in the text.

Changes: Lines 426-432, 544-553

**Comment 32**

Table 2: is there a specific reason for taking the absolute value of the accumulated difference? Leaving the sign would be more instructive (although it can of course be read from the PBIAS column).

Response: The absolute value of the accumulated difference will be added to the table and all plots.

Change: The accumulated difference has been added to table 1 and Fig. 6-13.

**Comment 33**

Section 5.2.1: I think the relevance of this section is minimal. The topic of the paper is on extreme precipitation, and the duration of precipitation is mostly determined by low-intensity precipitation. So I think most of this section (including Table 3) can removed, thereby improving the readability of this paper.

Response: The topic of the paper is not only extreme precipitation per se, but how the observations of such events differ between different types of sensors. Studying the duration of the same event observed by different sensors gives insights on 1) the sensitivity of the sensors 2) the temporal movement of the storm (which, as can be concluded from table 3, is from the south-west to the north-east in this case). In addition, we think it is an interesting finding that the CML and radars record similar durations. The duration also directly affects the calculation of return periods which we think is an important part of the discussion section. Furthermore, the evaluation metrics are computed for the duration of the event as recorded by each reference sensor (to exclude measurements of zero rainfall). For the given reasons, we suggest to keep this section in the paper.

Changes: None

**Comment 34**

Lines 440-442: a time lag of 25 minutes is necessary to optimize the correlation between the municipal gauge and the C-band radar. Such a lag is far beyond what would be expected from e.g. the time needed for the raindrops to fall from the radar measurement volume to the gauge. What can explain this difference. It is now stated the the C-band radar data should be shifted in time, but could it be that the clock on the municipal gauge was off? Please elaborate on this.

Response: After checking the calculations of time lags in detail they were found to be incorrect and we have now corrected the calculations.

Firstly, the event start at the municipal gauge was erroneously set to 17:15 in the calculations, when it in fact started at 17:05. As the metrics (and lags) are calculated for the duration of the municipal gauge when it serves as reference sensor, this will affect the calculation of time lags and metrics in Fig. 6 and Fig 8 as well.

Secondly, the radar time series were mistakenly sliced to the event duration of the municipal gauge before shifting them in time, meaning that some iterations was based on only a few time steps, including the optimum value (25 minutes) that was based on only 5 timesteps. Instead, the correlation should of course be calculated over the event duration (17:05-18:10 at the municipal gauge) while shifting the tested time series step by step. After this correction, the Spearman correlation is 0.4 for CWR compared with reference without time lag, which could be increased to 0.83 when shifting the CWR time series 10 minutes. We find this to be reasonable values.

Changes: Line 626, 654-655

**Comment 35**

Figures 6, 8, 9, and 12: please express the RMSE in mm h-1. This helps the reader compare to the y-axis that is used in the figures.

Response: RMSE will be expressed as mm/h in the figures.

Changes: RMSE expressed as mm/h in Figures 6, 8, 9, and 12.

**Comment 36:**

Lines 447-449: the duration of the event is not really relevant here. The gauge recorded 75.4 mm in 54 minutes, so it recorded the same amount of precipitation in 60 minutes. I would suggest removing this sentence, and mentioning that 75.4 mm was recorded by the gauge within 60 minutes here.

Response: The text will be updated to mention 60 minutes duration only.

Changes: Text updated in lines 635-636.

**Comment 37:**

Lines 450-451 and 457-459: the difference between the X- and C-band radars are very large, both in spatial structure and in total accumulation. Could beam blockage indeed explain this (see my point about adding information in Figure 1 before)? Or are there other reasons that could cause such underestimates and differences in spatial structure?

Response: We are confident that the difference in total accumulation is caused by beam blockage in this case, see responses to comment 14 and 47. The cause of differences in spatial structure is uncertain. The northeastern part of the study area has missing echo when comparing CWR with XWR – this might be a radar shadow behind the convective cell – but it is still very close to the radar location. Apart from this, it is hard to tell how much the two data sets should align given that they have different spatial resolution. It might have been better to resample XWR to 2km to allow for more consistent visual comparison with CWR.  If anything, these differences at least highlight the need for high-resolution data to observe convective storms.

2025-11-21
Louise Petersson Wårdh
louise.petersson.wardh@smhi.se

Changes: The comment has been addressed by adding appendix A1, and elaborating in the discussion (lines 786-796).

**Comment 38:**

Lines 464-465: the lag that optimizes the correlation between XWR and the municipal gauge time series is 10 minutes. This is very different from the 25 minutes reported for the C-band weather radar. Does that mean that there is a time difference between the clocks of the C-band and the X-band radars? Or could it be that the time lag that optimizes the correlation is not the true time lag. Assuming that both radar clocks are correct, I think that a single time lag should be determined based on the comparison with both radars.

Response: Following up on the response to comment #34, the same corrections were made for the XWR data. The Spearman correlation is 0.56 for XWR compared with reference without time lag, which could be increased to 0.7 when shifting the XWR time series 5 minutes. After these corrections we do not see a reason to believe that there is any significant difference between the clocks. Nevertheless, we will highlight the importance of accurate time recording in the discussion section.

Changes: Lines 815-819.

**Comment 39:**

Lines 466-469: in the discussion of Fig. 8 it is stated that the X-band radar underestimates the peak and overestimates the low intensities. I don't think you can draw that conclusion based on Fig. 8a. I see both over- and underestimation during heavy rain (i.e., everything above 10 mm h-1). And I see that the X-band radar reports quite heavy rain before and after the most intense part reported by the municipal gauge (17:10-18:10). Especially the peak just before 17:00 (more than 10 mm h-1) is significant (note that anything above 1 mm h-1 cannot be called drizzle). The gauge reports nothing at that time. I think a more thorough discussion of this figure should be included in the paper.

Response: We will rephrase as follows: "In Fig. 8a, there is a tendency that XWR underestimated the overall peak rainfall intensity and overestimated lower rainfall intensities". Note that we previously used the word "seems", which we do not think indicate a strong statement. The overall peak 17:30-17:40 recorded by the municipal gauge is indeed underestimated by XWR, and if we use the municipal gauge as reference, then the peak recorded by XWR at 16:55 when the gauge reports nothing is by definition an overestimation – the same holds for 18:15-18:35. However, there are indeed a couple of timesteps 17:45-18:10 when XWR reports higher intensities than the gauge, hence we will use the word "tendency".

Changes: Text updated in lines 657-658.

**Comment 40:**

Figure 9a: How is the spearman correlation coefficient computed given the fact that there are many CML data points that have exactly the same value (how do you determine the rank of these points)? I find it hard to believe that the correlation between CML and mean XWR is 0.9 given the large differences in the time series. Please check this number.

Response: Recalculation of the Spearman rank gives the same result. It is correct that Spearman rank should not be applied for time series with many multiple values (ties) but we have applied tiescorrection in the calculation (Emond et al., 2002). But as pointed out, it is probably also correct that the good metrics is a pure coincidence as the "plateau value" recorded by the CML happens to be in the middle range of the fluctuation of XWR observations, so the total accumulated depths are similar between CML and XWR reference. When computing the metrics for "non-plateau" values only the correlation is still 0.86 which suggest that CML and XWR record similar temporal variations during the storm. The absolute difference is then -15.5 mm, PBIAS 52 % and RMSE 4.1 mm/h for the non-plateau period, so a lot worse than with the "plateau-period" included. However, the non-plateau timesteps are outside the most intense part of the storm and therefore not the main focus of the analysis. We will keep the metrics as they are but use a more nuanced language, remove any formulations about that these metrics are "good" and discuss the fact that it is a coincidence in this case that the "plateau-intensity" leads to a similar total accumulation as the reference sensor.

Changes: Lines 676-678, 830-837

**Comment 41**

Figure 9b: when I look at the mean rainfall depth recorded by the X-band radar, and I compare this to the map in Fig. 7, these figures are inconsistent. Fig. 9 shows more than 60 mm, whereas the maximum rainfall depth on the link path in Fig. 7 is in the 50-60 mm class. Given the fact that parts of the link are in areas that received only 10-20 mm of rain according to Fig. 7, the numbers cannot be correct. The same holds for Fig. 8b where the graph indicates almost 80 mm of rain, but Fig. 7 shows between 50 and 60 mm. Please check the data behind Figures 7, 8, and 9, and correct where necessary.

Response: See response #26. The time series in Figure 8 and 9 are derived from XWR polar bins and Fig. 7 is gridded data which inevitably introduces averaging, especially for data with high spatial variability. This is explained in lines 455-457 but we will try to clarify this even further.

Changes: Some clarifications have been provided as described in response to comment #26.

**Comment 42**

Figure 9b: the gray area seem to be the accumulations based on the time series of the 10th and 90th But this is not really the spread in accumulations among the pixels over the CML. To show that, you would have to make the graphs of the accumulation for each pixel separately, and then take the 10thand 90th percentiles. I would expect that the spread would be more limited then (still significant, judging from Fig. 7). Please correct this in this figure.

Response: Thanks for pointing this out, we had accidentally plotted instantaneous spread instead of cumulative spread. The figure will be corrected accordingly.

Changes: Figure 9b updated with cumulative spread.

**Comment 43**

lines 482-483: my conclusion here would be that the fact that the metrics are so good is pure coincidence. So I strongly suggest to remove any conclusions about how good the CML performs for this case because you can't really conclude that based on the data.

Response: We agree and will update the text accordingly.

Changes: Lines 676-67

**Comment 44**

Figure 10: I really like this figure as it nicely demonstrates the enormous space-time variability of rainfall intensities for this storm and hence the challenge of sampling it well. Would it be possible to add the times (i.e., "17:22", "17:23", etc.) to each of the panels? That would increase the readability of the figure.

Response: Thank you. This was actually included in an earlier version of the figure and will be added again.

Changes: Time stamp added to the panels in figure 10.

**Comment 45**

Figure 11b: this negative correlation is expected if the exponent of the CML rainfall retrieval relation (a) as expressed in Eq. (A2) is smaller than 1 (see Leijnse et al., 2010; Eq (10); note that in this equation the exponent b of the retrieval relation is defined differently (b=1/a)). What is the value of a used in this paper?

Response: The value applied for alpha (for 23 GHz) is 0.96, i.e. b = 1.04. We will refer to Leijnse et al., (2010) to support this.

Changes: Lines 853-855, and lines 1040-1041 in appendix A2.

**Comment 46**

line 537: "expect" should be "except".

Response: Thank you, this will be changed.

Changes: Line 742

**Comment 47**

lines 579-581: it is suggested here that partial beam blockage may be the cause of the severe underestimation of the rainfall accumulations by the C-band weather radar. This could definitely be the case. However, there is not enough information in the paper to be able to conclude this. There are also other sources of error could have resulted in these underestimated, even at close range (see e.g. van de Beek et al., 2016). Please add the relevant information to the paper (the SMHI, 2020 reference doesn't give the relevant information).

Response: See response to comment 14. Based on internal evaluations and experience with this particular radar, we strongly believe that the siting of the radar (and consequently vegetation beam blockage) is the most contributing factor to the severe underestimation. See response #14 for more details. We will add a clause on other possible sources of the error and refer to the suggested paper.

Changes: Lines 787-796 and adding Appendix A1.

2025-11-21
Louise Petersson Wårdh
louise.petersson.wardh@smhi.se

**Comment 48**

lines 600-602: it is stated that the size of the bucket in the tipping bucket gauge (0.2 mm) could be the cause of the low correlation. However, given the 5-minute time sampling used for computing this correlation, a single tip per 5 minutes would correspond to 2.4 mm h-1. The intensities recorded by the X-band radar are much higher than that, even in the "calmer periods of the event", so this can't be a significant factor in the low observed correlation coefficient.

Response: Thank you for pointing this out, we will remove it as a possible cause for the low correlation with the municipal gauge.

Changes: Lines XX-XX have been deleted.

**Comment 49**

lines 608-611: the effect of partial beam blockage is discussed here. I don't think that near-ground observations are particularly prone to beam blockage as a rule. And the larger sampling volume of the X-band radar will not suffer less from beam blockage. The siting of the radar is a much more important factor in this. Please adapt the discussion in the paper to reflect this.

Response: Thank you for pointing this out, we will remove this sentence.

Changes: Lines 826-829 have been deleted.

**Comment 50**

line 612-613: I think that the statement that the CML correlated well with the X-band radar data is not something that should be highlighted here because of the plateau that the CML experiences (see also my earlier point about this).

Response: We agree and will update the discussion accordingly.

Changes: Lines 834-837 in Discussion

**Comment 51**

line 628: X-band variability is partly attributed here to attenuation, but it is mentioned in Section 3.2 (lines 189-190) that attenuation has been corrected for. Please elaborate on this.

Response: There might be local attenuation in the data even after application of attenuation correction as all methods have limitations and uncertainties. We will add a clause like "… due to possible uncertainties in the attenuation correction" or the like.

Changes: Lines 848-849

**Comment 52**

lines 632-634: the underestimation of both the C-band radar and CML with respect to the X-band radar are stated to be related to the shorter wavelength of the X-band radar. I don't think this is true. I strongly doubt that the wavelength of the CML is longer than that of the X-band radar (please

indicate the frequency of the CML in the paper). And given the very short range of the C-band radar and the typical sensitivity of operational weather radars, this radar should have no trouble in detecting even very light precipitation. So I don't think that the wavelength of the different sensors can explain the observed differences.

Response: It is correct that the frequency of the XWR (9.4 GHz) is of course lower than the CML (23.1 GHz) so this statement will be removed. Our investigations show that the CWR indeed has trouble at detecting light precipitation at this elevation angle (higher angles perform better). However, we think it is out of the scope of the paper to elaborate on this, so we will just delete this sentence.

Changes: Lines 856-858 have been deleted.

**Comment 53**

line 637: I think the differences in durations between the rain gauge on the one hand and the CML and radars on the other can't be attributed to wind drift. The other two reasons that are given seen very valid to me. So consider removing the wind drift as a possible cause for this.

Response: Thank you, wind drift will be removed as a possible cause for the observed discrepancy.

Changes: Line 861

**Comment 54**

lines 658-674: the station outlier quality control filter is discussed here. It doesn't seem to perform well for this storm, flagging valid data, and failing to flag data of questionable quality. You mentioned earlier (on lines 378-382) that you modified two of the parameters of the SO algorithm. To what extent do you think this has influenced these results? Please also discuss this here.

Response: The original values of the parameters $m_{match}$ and $m_{int}$ were 200 and 4032, which means that 200 rainy time steps in the last 2 weeks (for 5-minute data, which is used here) are required to compute Station Outlier flags (as we know, correlations should not be calculated on dry time steps). With these settings, all stations were flagged as "-1" ("filter cannot be applied") for the period of interest because there were not enough rainy timesteps in the last 2 weeks. This will be clarified in lines 379-380.  Instead, we require 100 wet timesteps in the last four weeks to apply the filter. The result of this is that no timesteps are flagged as "-1" in the period of interest but the SO algorithm can be applied at all time steps. The quality control is then based on less data (100 wet timesteps instead of 200) but it is a trade-off between having any quality control performed at all, and having it performed based on less reference data. So, the failure of the SO-algorithm cannot be attributed to the change of parameter settings but rather to the spatial variability of the storm. We will incorporate these reflections in the discussion section.

Changes: Added in line 895-898

**Comment 55**

lines 680-681: the excellent coverage of X-band radars in southern Sweden is discussed here. What fraction of the land surface of southern Sweden is actually covered by X-band radars? It would be interesting for readers to know this.

Response: We will change vague references like "southern Sweden" and "southwestern Sweden" to the more well-defined area "Skåne county" in section 1, 2, 3.2 and 6. The word "excellent" in the text refers to the combination of XWR and rain gauges operated by local authorities, not the XWR coverage itself. There are two X-band radars in Skåne (and all of Sweden) – their coverage is shown in Figure 1 in Hosseini et al (2023). This is approximately 9000 km$^2$, corresponding to 80 % of Skåne county and 2 % of all of Sweden.

Changes: "southwestern Sweden" changed to "Skåne County" throughout the text. Lines 219-221 added.

**Comment 56**

lines 687-688: rain/no rain detection is mentioned here. I don't think this is relevant for this paper (which is about intense convective precipitation). Furthermore, I don't think you can really conclude this based on the results presented here. It is very difficult knowing what the truth is with very light rainfall, given the sensors that have been used in this study. So I would suggest removing this from the paper.

Response: The intention of this statement is to highlight that third-party sensors can assist in understanding the spatial variability of a storm, while they may not always provide accurate information on intensity/depth at all points of interest. This phrasing is already used in the conclusion section and will be added here for clarity.

Changes: Lines 918-919.

**Comment 57**

Section 7: One thing that I think should be stressed more in the conclusions is that you've clearly shown that this storm is extremely variable in space (and that this holds for most storms that produce so much precipitation), and that all sensors that are added could potentially provide valuable information about parts of the storm that are otherwise not properly sampled. Could you stress this even more in the conclusions? This really shows why it is so important to continue research on second- and third-party data.

Response: Section 7 will be amended to put more emphasis on the added value of incorporating these sensor types as complementary observations in the context of extreme rainfall.

Changes: Conclusion section updated.

**Comment 58**

Appendix A: CML attenuation is discussed in this appendix, and it looks to me like attenuation (expressed in dB) and specific attenuation (expressed in dB km-1) are intermixed. Please correct any issues related to this, and include units whenever introducing a new variable.

Response: Thank you for noticing this, we will revise this part of the appendix and clarify the units.

Changes: Lines 1030-1034